# Mouse sensorimotor cortex reflects complex kinematic details during reaching and grasping

**Harrison A Grier[1], Sohrab Salimian[1], Matthew T Kaufman[2,3,4]\***

[1]Graduate Program in Computational Neuroscience, The University of Chicago, Chicago, United States; [2]Department of Organismal Biology and Anatomy, The University of Chicago, Chicago, United States; [3]Neuroscience Institute, The University of Chicago, Chicago, United States; [4]NSF-Simons National Institute for Theory and Mathematics in Biology, Chicago, United States

## eLife Assessment

The granularity with which neural activity in the sensorimotor cortex of mice corresponds to voluntary forelimb motion is a key open question. This paper provides **compelling** evidence for the encoding of low-level features like joint angles and represents an **important** step forward toward understanding cortical limb control signals.

**\*For correspondence:**
mattkaufman@uchicago.edu

**Abstract** Coordinated forelimb actions, such as reaching and grasping, rely on motor commands that span a spectrum from abstract target selection to detailed instantaneous muscle control. The sensorimotor cortex is central to controlling these complex movements, yet how the detailed command signals are distributed across its numerous subregions remains unclear. In particular, in mice, it is unknown if the primary motor (M1) and somatosensory (S1) cortices represent low-level joint angle details in addition to high-level signals like movement direction. Here, we combine high-quality markerless tracking and two-photon imaging during a reach-to-grasp task to quantify movement-related activity in the mouse forelimb M1 (M1-fl) and forelimb S1 (S1-fl). Linear decoding models reveal a strong representation of proximal and distal joint angles in both areas, and both areas support joint angle decoding with comparable fidelity. Despite shared low-level encoding, the time course of high-level target-specific information varied across areas. M1-fl exhibited early onset and sustained encoding of target-specific signals, while S1-fl was more transiently modulated around lift onset. These results reveal both shared and unique contributions of M1-fl and S1-fl to reaching and grasping, implicating a more distributed cortical circuit for mouse forelimb control than has been previously considered.

## Introduction

Primary motor cortex (M1) and primary somatosensory cortex (S1) are the neocortical structures most strongly and directly connected to the sensorimotor periphery. Accordingly, signals related to movement are strongly present in M1 (*Evarts, 1973*; *Galiñanes et al., 2018*; *Georgopoulos et al., 1982*; *Moran and Schwartz, 1999*; *Park et al., 2022*; *Saleh et al., 2012*; *Sauerbrei et al., 2020*), and signals related to both cutaneous and proprioceptive somatosensation are strongly present in S1 (*Alonso et al., 2023*; *Chowdhury et al., 2020*; *Costanzo and Gardner, 1981*; *Fromm and Evarts, 1982*; *Gardner and Costanzo, 1981*; *Pei et al., 2009*; *Prsa et al., 2019*; *Umeda et al., 2019*). Yet

despite being among the longest-studied brain structures, our understanding of these areas' exact roles in movement production remains incomplete (*Omrani et al., 2017*).

In both primates and rodents, M1 and S1 are known to be strongly interconnected (*Pons and Kaas, 1986*; *Huerta and Pons, 1990*; *Darian-Smith et al., 1993*; *Yamawaki et al., 2021*; *Muñoz-Castañeda et al., 2021*) and work closely together (*Nelson, 1996*): sensory signals are often strongly present in M1 (*Lucier et al., 1975*; *Lemon et al., 1976*; *Evarts and Fromm, 1977*; *Wong et al., 1978*; *Fetz et al., 1980*; *Fromm et al., 1984*; *Alonso et al., 2023*; *Asanuma, 1975*), and movement-related signals are often present in S1 (*Chowdhury et al., 2020*; *Cohen et al., 1994*; *Goodman et al., 2019*; *Nelson, 1987*; *O'Connor et al., 2021*; *Prud'homme and Kalaska, 1994*; *Tanji and Wise, 1981*; *Umeda et al., 2019*; *Weber et al., 2011*). Nevertheless, stimulation in M1 produces movement much more readily than in S1 (*Penfield and Boldrey, 1937*; *Tennant et al., 2011*), and lesions (*Lawrence and Kuypers, 1968*; *Whishaw et al., 1993*; *Kawai et al., 2015*; *Mizes et al., 2023*; *Mizes et al., 2024*; *Nicholas and Yttri, 2024*) and inactivations (*Galiñanes et al., 2018*; *Guo et al., 2015*; *Morandell and Huber, 2017*; *Miri et al., 2017*; *Hwang et al., 2019*; *Sauerbrei et al., 2020*; *Hwang et al., 2021*; *Park et al., 2022*) of M1 produce strong deficits in movement production, especially for movements that are kinematically complex (*Bollu et al., 2024*), flexible (*Heindorf et al., 2018*; *Mizes et al., 2023*), or coordinated across the hand and arm (*Guo et al., 2015*).

In monkeys, motor cortical activity has been argued to relate to high-level signals, such as movement endpoint (*Georgopoulos et al., 1986*) or hand velocity (*Kalaska et al., 1983*; *Ashe and Georgopoulos, 1994*; *Paninski et al., 2004*; *Wang et al., 2007*); or, low-level signals, such as joint kinematics (*Vargas-Irwin et al., 2010*; *Saleh et al., 2012*; *Saleh et al., 2010*; *Goodman et al., 2019*; *Okorokova et al., 2020*) or muscle forces (*Kakei et al., 1999*; *Lemon et al., 1986*; *Morrow and Miller, 2003*); or even very detailed control previously thought to be reserved for the spinal cord, such as selecting types of motor pools (*Marshall et al., 2022*). The presence of signals at all of these levels suggests that monkey M1 may play a role in multiple levels of control. However, it is challenging to tease these roles apart in a model system where fine-grained causal perturbations and circuit tools are less developed.

The mouse has emerged recently as a model of interest for dissecting motor control (*Warriner et al., 2020*). As a model system, it offers high throughput, detailed atlases through both anatomical and functional tracing (*Callaway et al., 2021*; *Muñoz-Castañeda et al., 2021*; *Winnubst et al., 2019*; *Zingg et al., 2014*), and powerful techniques for large-scale recording (*Manley et al., 2024*; *Steinmetz et al., 2021*; *Yu et al., 2021*) and perturbation (*Guo et al., 2015*; *Guo et al., 2021*; *Sauerbrei et al., 2020*). However, the study of systems-level motor control in actively behaving animals is less mature in the mouse than in primates.

Extensive motor mapping experiments in rodents have revealed that activating different parts of the sensorimotor cortex evokes movements of different body parts or different kinds of movements of the same body part, as it does in primates (for review, see *Harrison and Murphy, 2014*). Yet, it is unclear how the topography of stimulation-evoked movements relates to the roles of these areas during volitional actions. Perturbations during behavioral tasks in mice involving forelimb lever or reaching movements have provided a coarse-level understanding of how these areas contribute during behavior. Inactivations and lesions of M1 and S1 have shown that M1 is required for the execution of dexterous reach-to-grasp movements (*Guo et al., 2015*; *Sauerbrei et al., 2020*; *Galiñanes et al., 2018*; *Wang et al., 2017*; *Whishaw et al., 1991*; *Whishaw, 2000*) and that S1 is essential for adapting learned movements to external perturbations of a joystick (*Mathis et al., 2017*). However, spinal cord projections from mouse M1 and S1 primarily target spinal interneurons rather than directly synapsing onto motor neurons (*Gu et al., 2017*; *Wang et al., 2017*; *Ueno et al., 2018*), suggesting cortical activity might play a more modulatory role. Furthermore, stimulation of brainstem nuclei alone can evoke naturalistic forelimb actions, including realistic reaching movements involving coordinated flexion and extension of the proximal and distal limb (*Esposito et al., 2014*; *Ruder et al., 2021*; *Yang et al., 2023*). Taken together, these results have raised the question of what role mouse M1 and S1 play in the control of goal-directed forelimb movements.

One route to answering this question involves characterizing the signals present in mouse M1 and S1 during movement. If mouse M1 and S1 were to control only high-level aspects of forelimb movements, activity should be dominated by 'abstract' signals like target location and reflect little trial-to-trial variability in reach kinematics. If instead M1 and S1 control low-level movement features, then

activity should correlate strongly with forelimb joint angle kinematics and their trial-to-trial variation when reaching to different targets. While the presence of high- or low-level signals in a cortical area does not necessarily imply that they are causally responsible for these aspects of movement, characterizing what signals are present serves as a first step toward determining how these areas relate to movement.

Here, we take this step by characterizing what kinematic information is present in M1 and S1 during reaching and grasping. To do so, we combine a reach-to-grasp-to-drink paradigm (based on *Galiñanes et al., 2018*; *Galiñanes and Huber, 2023*) with detailed high-speed behavior tracking and two-photon calcium imaging to ask what movement-related signals are present in mouse M1 and S1. Decoding models revealed that both M1 and S1 encode fine details of movement in the mouse, enabling decoding of all 24 joint angles we reconstructed from the shoulder to digits – even features such as digit splay and opposition within the paw. Differences between the areas emerged only in the decoding of higher-level aspects of the task: M1 reflected target location earlier than S1, and S1 reflected movement initiation time more accurately than M1. Our results link mouse M1 and S1 activity to the execution of reach-to-grasp movements and highlight their distinct yet complementary roles in encoding low-level kinematic details and higher-order movement features.

## Results
### Complex movement kinematics during mouse reaching and grasping movements

Assessing the relationship between neural activity and the details of movement requires striking a balance between achieving repeatable behavior and evoking sufficient trial-to-trial variability to broadly sample movement space. To achieve this balance, we modified a previously developed water-reaching task (*Galiñanes et al., 2018*) to evoke challenging reaching and grasping movements to two locations (*Figure 1A*). Importantly, the task setup was made compatible with high-speed stereo camera imaging and modern markerless tracking techniques for tracking putative joint centers, allowing for a detailed quantification of reaching trajectories and grasp attempts. Using a custom 3D-aware frame-labeling GUI and large numbers of labeled frames, we achieved high-quality tracking on nearly every frame (99% of lift-locked frames passed quality assurance for all 24 joint angles; 94.5% of trials were tracked usably; see Methods). This allowed us to quantify the kinematics at a fine level of detail and to obtain a more comprehensive description of these movements through joint angle kinematics (see Figure 3).

Our task involved two water spout targets fixed in place on either side of the mouse's head at distances near the extent of the mouse's reach. After an initial hold period, the animals were instructed to reach for a specific spout by the pitch of a cue tone, which occurred simultaneously with water delivery at that spout. The probability of each spout being cued was 50% on each trial, so the animal had no information as to where they should reach until the cue was played. After the cue, they performed rapid reaching and grasping movements to retrieve the water reward from the spout (*Figure 1B*). The mice produced reach trajectories with substantial variability around the basic reach shapes (*Figure 1C*), likely resulting from small differences in initial posture causing execution errors and corrections. Three design choices increased this movement complexity: (1) the water spouts were located near the maximal range of motion of each animal, (2) the water spouts were designed so that water would cling to them, and (3) we required good contact between the paw and the spout using touch sensors. As a result, the mice often displayed repeated grasp attempts to contact the spout and retrieve the entire water droplet (*Figure 1G*). This variability across trials, spout targets, and mice was also seen in trial events like lift reaction time, reach duration, time-to-first-contact, and the number of contact events that occurred within 1 s of the cue on each trial (*Figure 1D–G*).

In similar tasks, optogenetic inactivation of mouse M1 can halt the ongoing execution of reaching and grasping movements (*Galiñanes et al., 2018*; *Guo et al., 2015*). However, given that many of our mice performed the task for upward of 8 weeks and evidence that motor cortex disengages after extensive practice with lever pressing (*Hwang et al., 2019*; *Hwang et al., 2021*; *Kawai et al., 2015*), it was not clear whether motor cortex would remain necessary in our task. To determine this, we performed optogenetic inactivations of the forelimb portion of M1 (M1-fl; *Muñoz-Castañeda et al., 2021*) by activating inhibitory neurons in two VGAT-ChR2 mice. These inactivations blocked

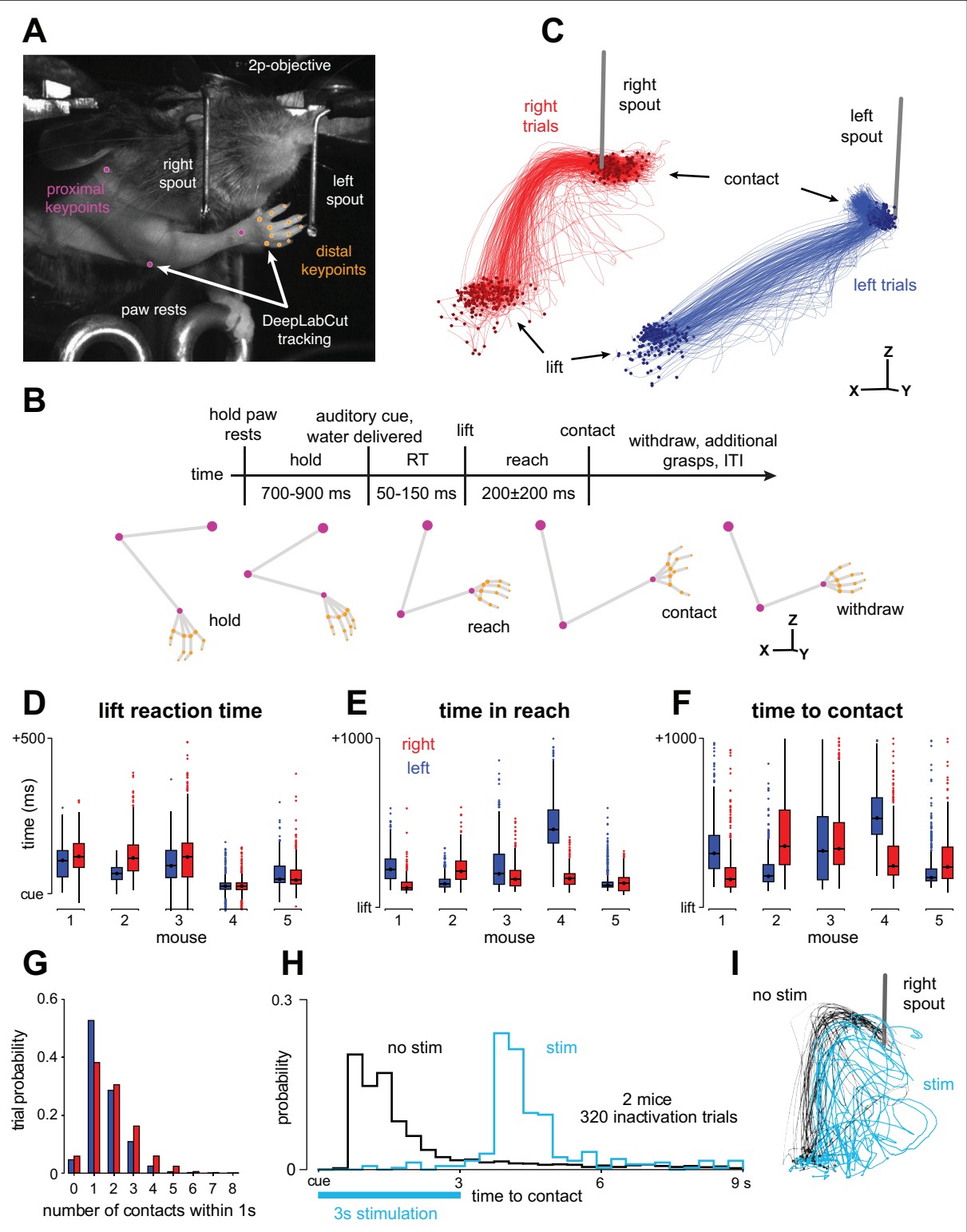

**Figure 1.** Mice make complex reaching and grasping movements to retrieve water rewards during head fixation. (**A**) Task setup: a head-fixed animal mid-reach with representative markerless tracking. (**B**) Task timeline, with representative 3D triangulated forelimb 'skeletons.' (**C**) Distal digit centroid trajectories from 100 ms before lift to 100 ms after contact. Stereo viewpoint corresponds to the data image in **A**, left and right trial traces offset for visualization. One example session. (**D**) Lift reaction time distributions, left and right trials depicted separately. (**E**) Time from lift to paw centroid arriving within 5 mm of the spout target tip. (**F**) Time between lift and spout contact. Box plots in **D**-**F** show the quartiles (box and central line), whiskers extend 1.5 times the interquartile range past the quartile boundary, and more distant points (outliers) are plotted individually. (**G**) Number of contact

*Figure 1 continued on next page*

*Figure 1 continued*

events within 1 s of the water cue per trial. Data were pooled across sessions for **D-G**. (**H**) Time between cue and spout contact during optogenetic experiments. (**I**) Representative digit centroid trajectories for a subset of right trials during optogenetic experiments, from –100 ms before lift to 1000 ms after lift. Data from one example session.

the execution of the reach to grasp sequence, preventing the animal from making contact with the spout during the 3 s laser stimulation period (*Figure 1F*; 86.5% control trials with contact within 3 s of cue, 5.1% inactivation trials with contact, *p*<10⁻¹⁹¹, Mann-Whitney U test, 2 mice, 495 stimulation trials). Interestingly, inactivation at the time of cue often did not prevent reach initiation (mouse 1: 54.7%, mouse 2: 34.2% of inactivation trials with lift within 3 s; 93.5%, 86.2% control trials). Yet the movement stalled once the paw and digits extended towards the spout, producing uncoordinated and unsuccessful reaching trajectories (*Figure 1I*, two representative datasets). Taken together, these results support the involvement of M1-fl in the water-reaching task and suggest that the strength of inactivation effects may depend on specific task details like training time or target configuration (*Galiñanes et al., 2018*).

## Stereotyped aperture kinematics across idiosyncratic reach-to-grasp movements

Rodent forelimb movements have been described qualitatively as a set of sequential stereotyped postures when reaching to grasp food or water rewards (*Whishaw and Pellis, 1990*; *Figure 1B*). We identified these features in our data using our high precision quantification of the mouse's proximal

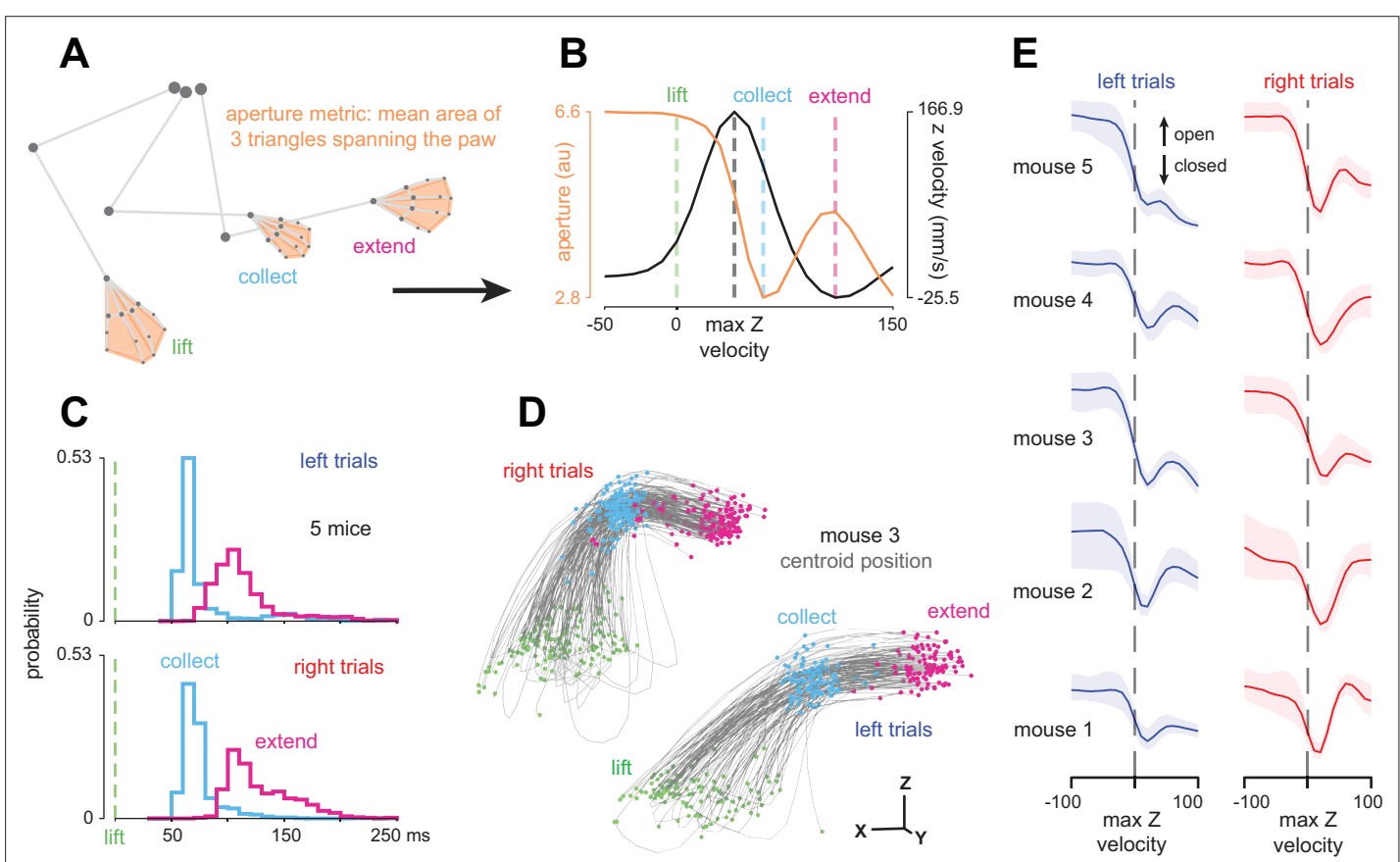

**Figure 2.** Paw aperture reveals stereotyped kinematics across idiosyncratic single-trial reach-to-grasp movements. (**A**) Paw aperture was computed from the paw markers using a simple geometric measure. (**B**) Aperture time series overlaid on the Z velocity of the paw centroid. The collect and extend events are defined by the aperture minimum and its subsequent local maximum. Representative trial. (**C**) Histograms of collect and extend event times for all trials, left and right trials separated. (**D**) Lift, collect, and extend event times depicted on digit centroid trajectories for one example mouse. Left and right trials offset in X and Z as in *Figure 1C*. (**E**) Median over trials of aperture time series aligned to the moment of maximum centroid Z velocity.

and distal forelimb during reaching. First, we defined a simple geometric measure of paw aperture that captured the strongest features of paw opening and closing, including the splay of the digits and flexion of the metacarpophalangeal (MCP) and proximal interphalangeal joints (PIP) (*Figure 2A*). The first aperture minimum corresponded tightly with the peak of vertical paw velocity after lifting, with the paw reaching maximum vertical velocity shortly before the paw and digits became maximally flexed (*Figure 2B*) ($r$=0.6–0.95 across mice, Pearson correlation; maximum velocity preceded minimum aperture by 27.2±10.5 ms across mice). When aligning single-trial aperture traces to the peak vertical velocity (max Z velocity), we observed clear, stereotyped structure in how paw aperture evolved during the movements, across both spout targets for all five mice (*Figure 2E*). This stereotyped shape was surprising given the heterogeneity in reach trajectories in this task and the strong differences in reaching strategies adopted for each target.

We termed the first aperture minimum after the lift to be the time of 'collect,' following *Whishaw et al., 2010*, and termed the next maximum after collect 'extend' (*Figure 2B*). These events structured the elaboration of the reach-to-grasp movement over time: after lifting, the paw closed to its minimum aperture (collect) before the digits were maximally opened (extend) as the paw was moved towards the spout. Extend occurred more heterogeneously in time after collect (*Figure 2C*), perhaps reflecting single-trial variability in initial posture. These grasp-related events corresponded strongly with the reach's progress as well. Regardless of the initial paw location on the paw rest, lifting the arm brought the flexing paw to a consistent location at the moment of collect, from which the mouse could prepare an accurate extension movement towards the spout (*Figure 2D*). These observations serve as quantitative confirmation of previous qualitative details obtained from high-speed video and confirm rich structure in the mouse reach-to-grasp movement.

## Proximal and distal joints are coordinated during mouse reach-to-grasp

To obtain a high-fidelity representation of the reach-to-grasp kinematics, we constructed a 3D 'skeletal' representation of the forelimb by video-tracking keypoints at 15 putative joint centers along the proximal and distal axis (Methods). We then used axis-angle geometries to approximate 24 joint angles (7 proximal, 17 distal), including a number of intrapaw angles that captured deformations of the paw itself (*Figure 3A*). These joint angles displayed rich temporal heterogeneity when reaching and grasping to both targets (*Figure 3B*). Reaching evoked particularly large rotation of the shoulder, likely because the mice reached from a quadrupedal position to targets on either side of the snout. Joint angles were strongly correlated across the entire limb, consistent with the expectation that the forelimb is a system of joints that are controlled cooperatively and, in some cases, are mechanically linked (*Figure 3C*). The strongest correlations occurred for angles describing the same joint (e.g. the three angles describing the shoulder) and for the flexion of MCP and PIP joints. The block-like structure of the joint angle correlation matrices indicated that control of the mouse distal limb is tightly linked to control of the proximal limb, consistent with previous observations of the strong stereotypy in mouse forelimb movements across behaviors (*Naghizadeh et al., 2020*; *Whishaw, 1996*; *Whishaw et al., 2010*; *Whishaw and Pellis, 1990*).

Despite the strong coupling between the proximal and distal joint angles, rich variation remained in the action of different joints over time. The presence of strong correlations across joints suggested that the kinematics may be well described by a smaller number of independent degrees of freedom than the total number of recorded angles. To assess the number of linearly independent (uncorrelated) degrees of freedom amongst the 24 joint angles and velocities, we used double-cross-validated PCA (*Yu et al., 2009*; Methods; *Figure 3D*), finding intermediate dimensionalities of 7 (median for joint angles) and 10 (velocities; *Figure 3E*). This is consistent with the idea that joint angles across the limb are coordinated instead of controlled independently, and that this coordination is flexible enough over time to enable accurately performing reaching and grasping to different targets.

## Layer 2/3 cells in M1-fl and S1-fl are strongly modulated and heterogeneously tuned

Having characterized the structure of the mouse forelimb reaching and grasping movements, we next sought to determine how the kinematic details of these movements were distributed across the sensorimotor cortex. To do so, we recorded neural activity from neurons in layer 2/3 of M1-fl extending slightly into the immediately adjacent secondary motor cortex and the forelimb region of S1 (S1-fl)

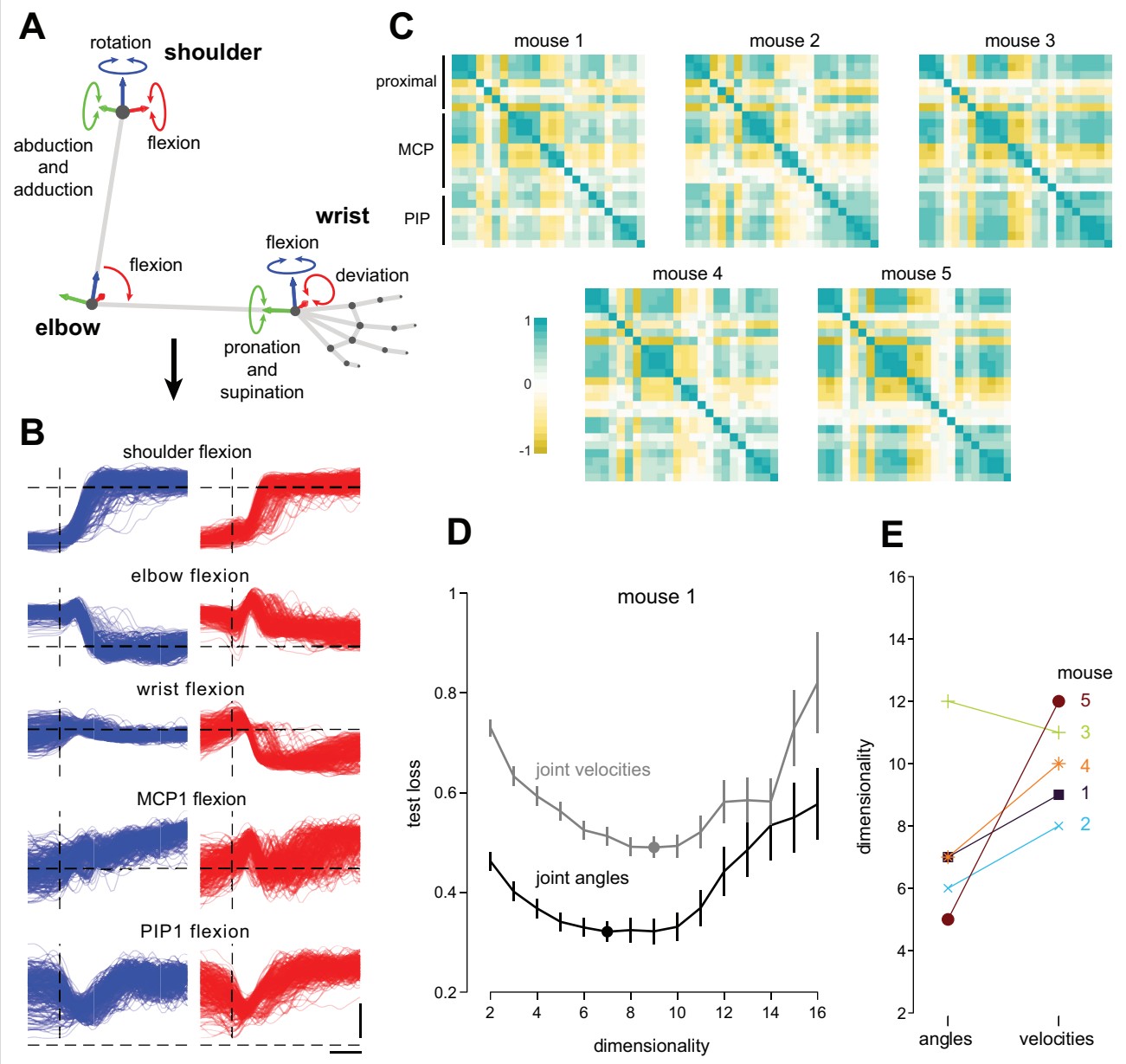

**Figure 3.** Mouse reaching and grasping movements involve coordination across proximal and distal joints. (**A**) Schematic of Euler angle inverse kinematics procedure, depicting seven joint angles over the proximal joints; 17 distal angles not shown. (**B**) Time series of 5 representative joints, shown from –100 to +400 ms around lift onset (vertical dashed line). Left (blue) and right (red) trials are shown separately. The horizontal scale bar is 100 ms. The vertical scale bar is 30 degrees. Dashed horizontal lines are 0 degrees. (**C**) Correlation matrices of joint angles over all recorded time points. Rows are ordered descending from proximal (shoulder) to distal (metacarpal phalanges, MCP, proximal interphalanges, PIP). (**D**) Example loss curve of the cross-validated principal component analysis performed on joint angle and joint velocity time series. (**E**) Estimated dimensionalities of joint angle and velocity kinematics across mice.

using two-photon calcium imaging of GCaMP6f-expressing neurons in layer 2/3 (185–230 µm deep, imaged at 31 Hz, cells extracted with Suite2p; *Pachitariu et al., 2017*). A slight majority of recorded neurons in both areas (51.8% in M1-fl and 52.2% in S1-fl, p<0.05 with Bonferroni correction for six tests) were statistically modulated to at least one event during the task: cue, paw lift, and/or first spout contact (*Figure 4B*; see Methods, 'ZETA'). Cells showed strong modulation locked to different task events, such as the cue onset (51.7%/52.7% in M1-fl/S1-fl, left or right *p*-value <0.05 corrected for dual tests), paw lifting from the rest (51.6%/54.8% in M1-fl/S1-fl), and time of first contact with the spout (45.2%/46.4% in M1-fl/S1-fl).

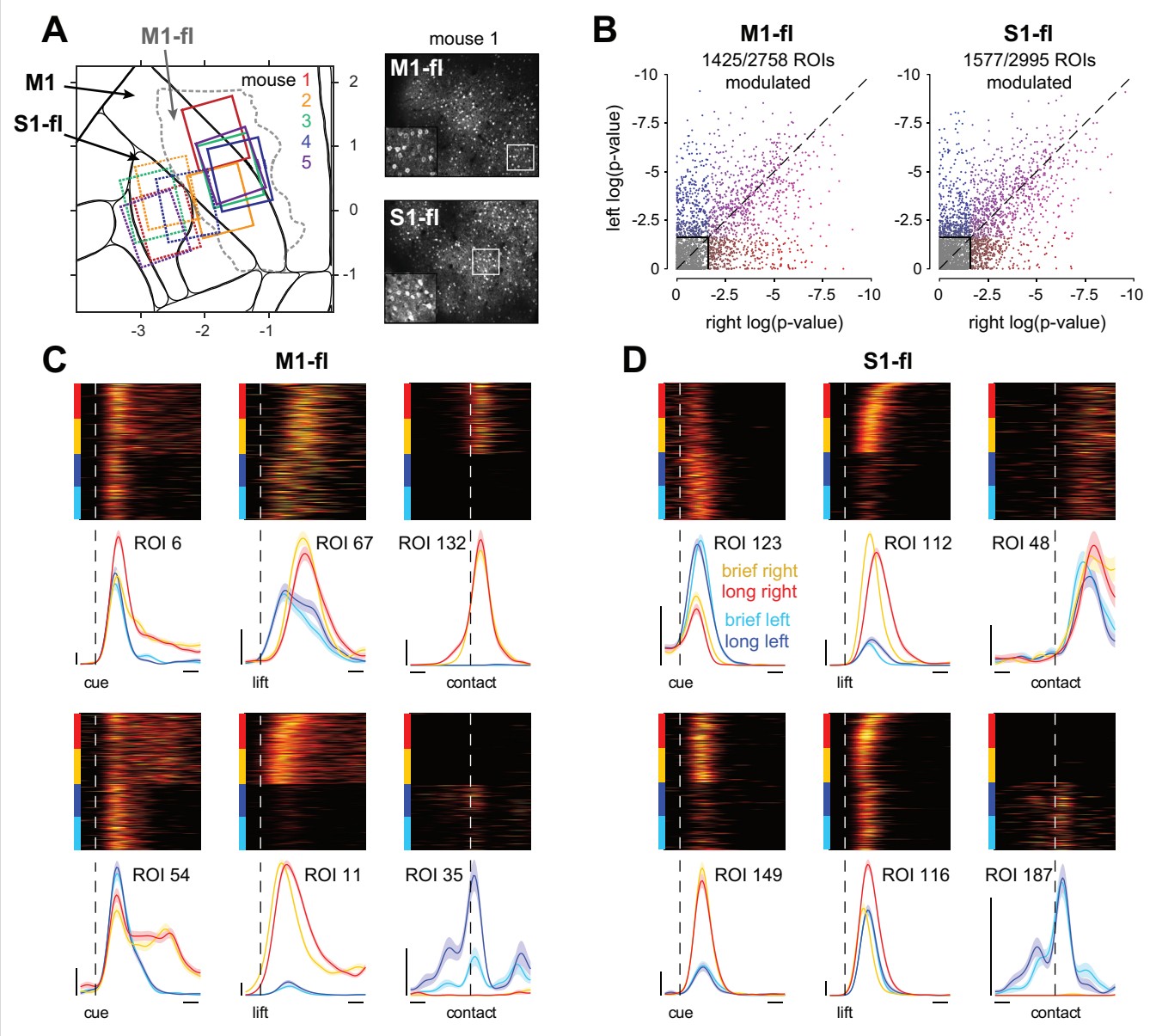

**Figure 4.** M1-fl and S1-fl cells are heterogeneously tuned to specific movement features. (**A**) Two-photon imaging fields of view for all mice aligned to the Allen Atlas common coordinate framework (CCF). Primary motor (M1) corresponds to MOp in the CCF, and S1-fl corresponds to SSp-ul. The dashed gray line indicates the outline of the forelimb primary motor cortex (MOp-ul, referred to here as M1-fl) as described in *Muñoz-Castañeda et al., 2021*. See *Figure 4—figure supplement 1* and Methods for field of view alignment details. Right, example max-projection images for M1-fl and S1-fl for mouse 1 with magnified insets. (**B**) Modulation metric (see Methods) of cue-locked data for individual cells in M1-fl and S1-fl. Values displayed are the logarithm of the computed p-value, where each point represents a cell. All cells with *p*-values <0.025 are shown colored: left-only modulated cells are blue, right-only modulated cells are red, and cells modulated to both conditions are purple. (**C**) Peri-event histograms and rasters of single-trial activity from individual cells in M1-fl. Each column displays two cells locked to each trial event: cue, lift, and first-spout contact. Trials grouped within each condition by time in reach. Vertical scale bar: 10 events per second (arbitrary units). Horizontal scale bar: 100 ms. (**D**) As in **C** but for S1-fl.

The online version of this article includes the following figure supplement(s) for figure 4:

**Figure supplement 1.** Vibration mapping for each individual mouse.

Cells were also modulated by the time taken to complete the reach: many cells exhibited different firing rates on trials with movement duration briefer versus longer than the median (*Figure 4C–D*, see ROIs 35 and 116). Finally, many cells showed temporal heterogeneity in their activity beyond that of simple transient bumps. This included cells (such as ROI 54) that were strongly modulated by movement onset and remained active late into the trial. This activity may reflect the details of movement as

the animals made repeated grasping attempts with a fully extended arm to retrieve the entire water reward from the spout.

## Proximal and distal joint kinematics can be decoded from M1-fl and S1-fl population activity

To assess how movement-related signals were distributed across M1-fl and S1-fl, we decoded the time series of each joint angle from neural population activity in each area. A simple linear regression model related the single-trial joint angles at all time points to single-trial neural activity at the corresponding moments. The model was fit with ridge regression, the ridge penalty was determined via a heuristic (*Karabatsos, 2018*), and performance was measured on held-out trials (90/10 train/test split, 50 folds). We found that every measured proximal and distal joint angle could be decoded from M1-fl activity at above-chance levels during reaching and grasping (*Figure 5A–B*). Interestingly, S1-fl decoding exhibited similar decoding performance as M1-fl across all mice (*Figure 5E–F*). Decoding was also possible with milder processing, or of joint angle velocities (*Figure 5—figure supplement 1*). Including additional causal (100 ms preceding) and/or acausal (100 ms preceding to 100 ms following) lags improved decoding performance modestly and similarly for both areas (*Figure 5—figure supplement 3E-F*).

Across both areas, we found that proximal joints were better decoded than distal joints (*Figure 5D and H*). A number of reasons for this difference are possible: this may reflect the time scale of 2p-recorded neural activity, overall smaller total variance of the most distal joint angles (see *Figure 5—figure supplement 2E–F*), or measurement noise present in the markerless tracking for small features like the fingertips. Nevertheless, we were able to explain substantial variance in the most distal intrapaw joints (see joints MCPf, MCPa, MCPo, PIPs, and PIPf).

Additionally, we reconstructed the skeletal representation of the forelimb from the decoded joint angles and found that, as intended, the reconstructed postures had strong qualitative resemblance to the true postures, even of 'minor' angles like cylindrical paw deformation (opposition) or digit splay (*Figure 5C and G*). This suggests that movement-related signals that could be used to actuate forelimb joints are present in both M1-fl and S1-fl populations.

Finally, we tested whether the ability to decode these many joint angles was a direct consequence of inter-joint correlations and might not be indicative of the presence of 'real' information about some of these joints. To do so, we fit partial correlation models that removed correlations between proximal and distal joints, or removed correlations of the joint angles with a high-level parameter – the overall distance of the paw centroid to the spout. Despite substantially lowering the behavioral variance, in each case, the residuals could still be decoded from neural activity (*Figure 5—figure supplement 2A–D*). Similar decoding performance for M1-fl and S1-fl was obtained from models fit to decode single-trial residuals separately for left and right trials (*Figure 5—figure supplement 3*), indicating that trial-to-trial variations on each basic movement were decodable from these populations.

Overall, these results reveal that neural activity in M1-fl and S1-fl is closely related to the kinematic details of reach-to-grasp movements. The ability to decode substantial variance in proximal and distal joints suggests that this relationship extends to the entire forelimb, and the similar performance obtained from each area suggests that this information is similarly distributed across M1-fl and S1-fl.

## M1-fl reflects target-specific information earlier and more persistently than S1-fl

Given the similar levels of kinematic information in M1-fl and S1-fl, we sought to determine if these areas differed in their encoding of a higher-level aspect of movement, target location. To assess how target-related signals were organized in each area, we analyzed how well target identity could be decoded from neural activity over time. For each session, a linear classifier was trained to predict whether the neural activity from a particular window of time came from a left-target or right-target trial. This decoder was then applied to held-out trials from the same time window. This analysis revealed that M1-fl activity reflected the target location slightly more faithfully than did S1-fl, as shown by its higher overall performance (93.2% M1-fl vs. 89.1% S1-fl median over cross-validation folds and time bins; $p=4.1\times10^{-4}$, Wilcoxon rank sum test). To examine single trials, we then projected each trial's population activity onto the identified decoding axis (the 'target-specific dimension'). In this projection, M1-fl activity predicted the target sooner after the cue than did S1-fl activity (59.9±34.7

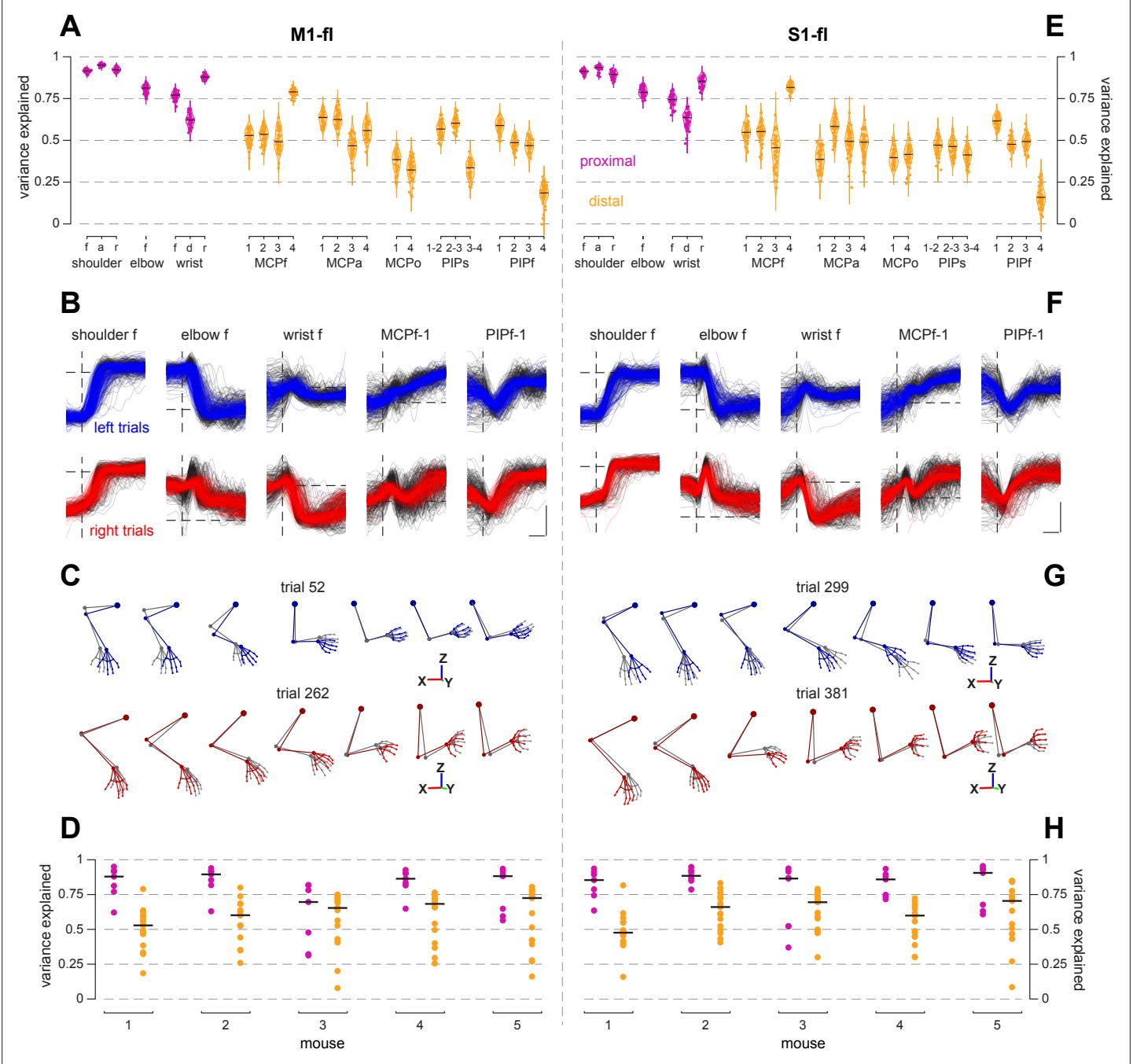

**Figure 5.** Kinematic details of proximal and distal joints during reaching and grasping can be decoded from M1-fl and S1-fl population activity. (**A**) Performance of decoding models predicting joint angle kinematics from RADICaL-inferred M1-fl rates. Violins depict the distribution of variance explained over cross-validation folds for each individual joint angle. (f: flexion, a: abduction/adduction, r: rotation, d: deviation, o: opposition, s: splay, MCP: metacarpal-phalanges, PIP: proximal-interphalanges; digits indicated by number). (**B**) Reconstructed joint angle time series for 5 representative joint angles. Black traces are original and colored are reconstructed; left trials are blue and right trials are red. A vertical dashed line indicates lift time, and a horizontal dashed line indicates 0 degrees. The vertical scale bar is 30 degrees, and the horizontal scale bar is 100 ms. (**C**) Forelimb skeletal postures reconstructed from decoded joint angles for representative left and right trials (trials having median variance explained). Seven representative time points from lift time to grab time are shown. Gray postures are original data, colored postures are reconstructed as in **B**. Decoded joint angles in each time step were applied to the neutralized posture of the original data, preserving the original lengths of inter-marker links (**D**). Performance of decoding models for all 5 mice. Each point is the median of the distribution of variance explained over cross-validation folds, with proximal (magenta) and distal (goldenrod) angles considered separately. The black lines indicate the median variance explained over all proximal or distal joints for an individual mouse. (**E–H**) As in **A-D** but for S1-fl data.

*Figure 5 continued on next page*

*Figure 5 continued*

The online version of this article includes the following figure supplement(s) for figure 5:

**Figure supplement 1.** Simpler processing or decoding joint angle velocities also produce strong kinematic decoding.

**Figure supplement 2.** Decoding of proximal and distal joint angles is not due to correlations between them or correlations with distance to target.

**Figure supplement 3.** Decoding the variation around a single target is possible, and adding lags slightly improves decoding.

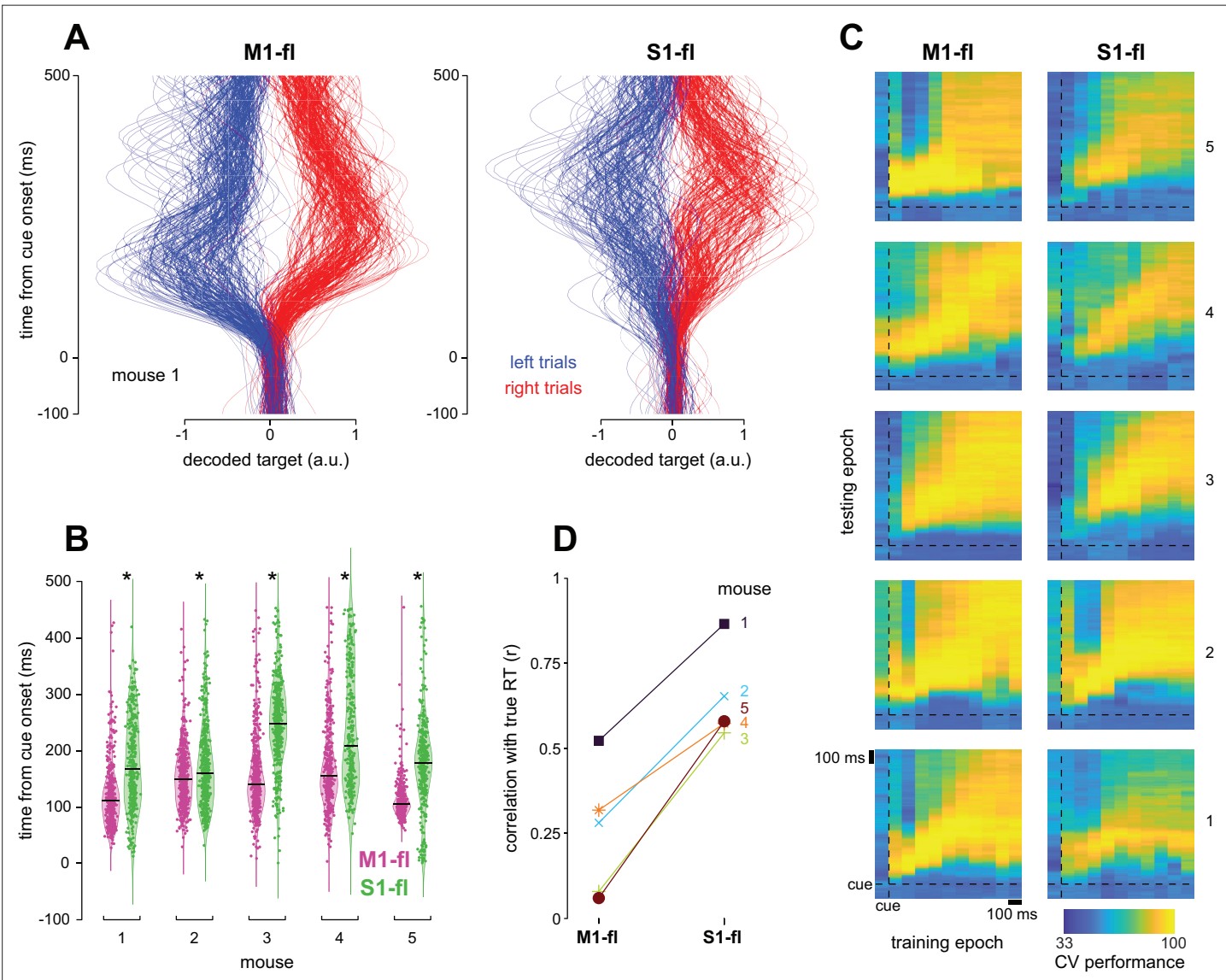

**Figure 6.** M1-fl reflects target-specific information earlier and more persistently than S1-fl. (**A**) Projections of population activity from held-out single trials onto the target decoding dimension for M1-fl and S1-fl. Right trials are red and left trials are blue. Traces were normalized by the 90th percentile value across trials for left and right trials separately. (**B**) Distribution of threshold crossing times for projections shown in **A**. Black lines indicate medians. Star indicates significantly different medians at $p<0.05$, Wilcoxon ranksum. (**C**) Performance of time-generalized target decoding analysis. Each heat map depicts the performance over all model training windows (columns) and test time points (rows). Time in trial runs vertically as in **A**. Both the training and testing epochs were −100 (leftmost column training, bottommost row test) to +1000 ms (rightmost column training, top row test) relative to cue. Training epochs were 100 ms non-overlapping windows and testing epochs were 10 ms bins. (**D**) Performance of movement onset prediction for M1-fl and S1-fl. The correlation coefficient r was computed between the true and predicted reaction times (RTs) across held-out trials.

The online version of this article includes the following figure supplement(s) for figure 6:

**Figure supplement 1.** Target classifier performance over time.

ms earlier, $p<0.05$ for all mice, Wilcoxon rank sum test) and displayed a stronger, more consistent, and longer-lasting prediction throughout the trial (*Figure 6A and B*).

Target decoding performance could result from truly higher-level signals that code abstractly for target location, or alternatively could be supported by strong encoding of kinematic variables that differed between targets. To disambiguate these possibilities, we refit the linear classifier to neural data after regressing off variance related to the joint angle kinematics. The strength and exact time course of the resulting target decoding varied somewhat across animals, but the earliest portion of target decoding performance persisted in all animals after the removal of kinematics, and performance remained stronger for M1-fl than S1-fl (*Figure 6—figure supplement 1B*). We thus conclude that higher-level signals are present in both areas, but differ in their exact timing and strength. However, we note that other possible signals, such as postural changes, could not be controlled for here.

To determine whether the target-specific dimension was conserved or whether it changed over the course of a movement, we then tested how well a decoder trained during one time window generalized to held-out data from other time windows. M1-fl decoders were more stable over the trial and performed better when generalizing to far-away time points than did S1-fl decoders (*Figure 6C*). These results suggest that the encoding of target-specific information is more consistent over time in M1-fl than S1-fl.

Finally, we asked how strongly each area encoded a different high-level parameter, resting versus reaching. To do so, we trained a decoder to predict whether neural activity was from a time point pre- or post-lift, then computed a movement onset time from when the decoder switched from pre- to post- (Methods; *Figure 6D*). Unlike our results for predicting target identity, S1-fl activity more strongly predicted lift time than did M1-fl ($p=0.0079$, Wilcoxon rank sum).

## Discussion

The extent to which activity in sensorimotor cortex reflects the details of movement has remained unclear in the mouse. Here, we show that in a challenging reach-to-grasp task evoking variable movements, both M1-fl and S1-fl relate tightly to all the joint angles we could track throughout the arm, paw, and digits. Using high-fidelity markerless tracking, we replicated prior, coarser results that the reach-to-grasp kinematics have stereotyped structure and involve strong coordination across proximal and distal joints. Using two-photon calcium imaging, we revealed that single neuron activity in M1-fl and S1-fl is strongly modulated and tuned to fine movement features. Decoding models then demonstrated that M1-fl and S1-fl both possess rich representations of low-level joint angle information at similar levels of fidelity. Despite these similarities, target decoders revealed that high-level movement information was stronger and more persistent at the population level in M1-fl than S1-fl. Activity in S1-fl was instead more transient and more closely linked to lift onset than in M1-fl.

These results expand our understanding of the rodent sensorimotor system and highlight similarities to nonhuman primates. We show here evidence in mice of detailed joint angle kinematic signals from the full forelimb in M1 and S1, as has been shown in macaque cortex during tasks involving reaching and grasping objects (*Vargas-Irwin et al., 2010*; *Saleh et al., 2010*; *Saleh et al., 2012*; *Goodman et al., 2019*; *Okorokova et al., 2020*). Additionally, the earlier onset of movement-related activity in M1-fl compared to S1-fl is similar to macaque M1 and S1 (*Tanji and Evarts, 1976*). Taken together, these results suggest that the mouse can be employed to address questions traditionally explored in primates about how cortical activity encodes detailed movement commands.

The strength of movement-related activity in these areas is surprising given the results of loss-of-function experiments in rodents, which have suggested that sensorimotor cortex can become disengaged from forelimb control with extensive training (*Hwang et al., 2019*; *Hwang et al., 2021*; *Kawai et al., 2015*), and that subcortical structures like the medulla or basal ganglia can independently control coordinated forelimb movements (*Esposito et al., 2014*; *Ruder et al., 2021*; *Yang et al., 2023*). However, several other previous results have indirectly suggested that M1 and S1 may be involved in the details of forelimb movement. Performance suffers with inactivation or lesioning of M1 and S1 in skilled, complex manual behaviors (*Guo et al., 2015*; *Mizes et al., 2024*; *Whishaw et al., 1991*) or idiosyncratic use of digits to accomplish non-dexterous tasks (*Kawai, 2014*). The sparing of non-dexterous tasks with these lesions may also reflect redundancy in control as opposed to irrelevance of M1 and S1. Nevertheless, our finding of low-level kinematic information in sensorimotor

cortex supports a role for cortex beyond simply providing redundant high-level commands to these subcortical areas.

There are a number of possible roles for this movement-related cortical activity. One is that the motor cortex may dominate control when the behavior is complex enough in some way – like when a cue is needed to identify the target (*Mizes et al., 2024*) or when sensory feedback is necessary to coordinate dexterous grasps in the presence of errors (*Guo et al., 2015*; *Perich et al., 2024*; *Sauerbrei et al., 2020*). Despite the overtraining of our behavior, these task features might preclude use of the presumably less-flexible subcortical circuits alone. Alternatively, subcortical areas like the medulla or cerebellum may initiate the movement (*Dacre et al., 2021*; *Esposito et al., 2014*; *Inagaki et al., 2022*; *Ruder et al., 2021*) while the cortex elaborates the details of otherwise-generic commands (*Kaufman et al., 2016*). This might permit the previously observed disengagement of cortex with learning of simple behaviors, could explain our observations of partially executed reach-to-grasp movements during M1-fl inactivation, and would explain the weak representation of lift-onset in our M1-fl data. A third alternative is that the medulla alone executes low-level control, but low-level information is maintained in the sensorimotor cortex for some other function, such as to enable learning or error correction (*Kawai et al., 2015*; *Makino et al., 2016*; *Wolff et al., 2022*; *Kirk et al., 2024*; *Koh et al., 2025*). These hypotheses are not mutually exclusive.

Our observation of detailed kinematic signals in the mouse forelimb somatosensory cortex raises the possibility of a central role for S1-fl in dexterous movements and may implicate a more broadly distributed circuit in forelimb control than previously considered. Indeed, the magnitudes of representation in M1-fl and S1-fl were nearly identical. This may simply be a consequence of widely distributed representations of movement across mouse cortex (*Musall et al., 2019*; *Steinmetz et al., 2019*; *Stringer et al., 2019*), including forelimb somatosensory areas, or may be a consequence of the close physical proximity of M1-fl and S1-fl hindering development of functionally distinct representations (*Tennant et al., 2011*). However, the strength of detailed kinematic information in S1-fl supports a role beyond simply receiving a broad corollary discharge signal.

One possibility is that proprioceptive signals are strong enough to carry these movement details into S1-fl from subcortical structures, as is true in primate forelimb S1 (*Umeda et al., 2019*). Our lift time decoding results are consistent with this view and align with recent observations characterizing mouse proprioceptive forelimb cortex (*Alonso et al., 2023*), although an alternative explanation may be simply that M1-fl activates earlier than S1-fl during reaching (*Kargo and Nitz, 2004*; *Miri et al., 2017*; *Veuthey et al., 2020*). Furthermore, our task evoked repetitive grasping attempts, which may engage S1-fl strongly to inform motor areas of the sensory consequences of movement (*Perich et al., 2024*). Extensive circuit tracing has shown that forelimb M1 and forelimb S1 are highly interconnected in the mouse (*Muñoz-Castañeda et al., 2021*; *Yamawaki et al., 2021*), suggesting that detailed movement signals in S1-fl may reflect not only sensory feedback but also efference copies of motor commands generated by M1-fl. Rodent S1 is known to influence interneuron populations in the spinal cord through direct and indirect projections that predominantly target the dorsal horn (*Ueno et al., 2018*), and thus these signals may also reflect S1-fl's important role in modulating reflex circuits to coordinate sensory feedback with movement generation (*Moreno-López et al., 2016*; *Moreno-Lopez et al., 2021*; *Seki et al., 2003*).

Several limitations of the present work should be noted. Pose estimation was performed with state-of-the-art neural networks (*Mathis et al., 2018*) and extensive hand-labeling of keypoints together with quality assurance metrics, but behavior tracking is inevitably imperfect. In particular, a noise floor remains that particularly impacts small bones, such as the distal phalanges, and markers estimated at the skin surface cannot perfectly reflect underlying joint and muscle kinematics. Additionally, calcium imaging has limited temporal resolution and is a nonlinear transformation of spiking activity (*Stringer and Pachitariu, 2019*; *Wei et al., 2020*). Although we used leading-edge deconvolution methods (*Friedrich et al., 2017*) and cutting-edge neural network approaches for sub-frame event estimation (*Zhu et al., 2022*), these limitations cannot be entirely overcome. These slow temporal dynamics limited our ability to disambiguate whether joint angles or joint velocities are more linearly encoded in these areas. The use of calcium imaging further prevents strong conclusions about whether activity reflects future limb states or sensory consequences. Confirming this limitation, inclusion of lagged data in the decoding models, whether causal or acausal, resulted in similar performance changes in both areas. Moreover, as is widely known (*Todorov, 2000*), the exact role of these kinematically-related

signals is challenging to determine from correlative measures alone; thus, determining whether these signals are used for direct movement control or indirectly reflect control performed elsewhere is a topic for future work. Nevertheless, our ability to decode substantial variance from small joint angles and fast changes within the paw supports the claim that movement signals are represented in M1-fl and S1-fl activity. Finally, all recordings included here were taken from layer 2/3 of both M1-fl and S1-fl. Activity may differ in layer 5, where the primary subcortical outputs originate.

Overall, our results suggest that mouse sensorimotor cortex is intimately involved in the control of dexterous forelimb movements and support use of the mouse as a first-class model system for studying the cellular and circuit basis of motor control. The presence of detailed kinematic signals in the sensorimotor cortex supports a model of mouse sensorimotor cortex in which M1-fl and S1-fl play a role in shaping the fine details of reaching and grasping movements. Further work will be required to map the roles of other sensorimotor cortical areas, understand their interactions, and differentiate between ballistic reaching and feedback-driven grasping and withdrawal.

## Methods

### Animal subjects

All procedures were approved by the University of Chicago Institutional Animal Care and Use Committee. A portion of the data presented here were analyzed in a previous study (mice 1 and 2; *Zhu et al., 2022*). For two-photon imaging, five male Ai148D transgenic mice (TIT2L-GC6f-ICL-tTA2, stock 030328; Jackson Laboratory) aged 8–12 weeks were used. Four of the mice carried the GCaMP6f transgene heterozygously and one homozygously (mouse 3). For optogenetic experiments, two VGAT-ChR2 (stock 014548; Jackson Laboratory) transgenic mice aged 8–12 weeks were used. Each animal underwent a single surgery. Mice were then individually housed in a reverse 12 hr light/dark cycle, with an ambient temperature of 22°C and a humidity of 58%. Experiments were conducted in the afternoon, during the animal's dark cycle.

### Surgical procedures

Mice were injected subcutaneously with dexamethasone (8 mg kg$^{-1}$) 24 hr and 1 hr before surgery to reduce inflammation. Mice were anesthetized with 2–3% inhaled isoflurane gas and then injected intraperitoneally with a ketamine–medetomidine solution (60 mg kg$^{-1}$ ketamine and 0.25 mg kg$^{-1}$ medetomidine). Mice were given supplemental low-level isoflurane (0–1%) if they showed any signs that the depth of anesthesia was insufficient or began emerging from anesthesia towards the end of the surgery. Animals received a subcutaneous injection of meloxicam (2 mg kg$^{-1}$) at the beginning of the surgery and once daily for 2 days post-operation. The scalp was shaved, cleaned, and resected, then the skull was cleaned and the wound margins glued to the skull with tissue glue (VetBond, 3 M). A circular craniotomy was then made with a 3 mm or 4 mm biopsy punch centered over the left MOp / SSp-ul border, as defined by the Allen Common Coordinate Framework v3 (Allen CCF). The coordinates for the center of M1-fl were taken to be 0.4 mm anterior and 1.5 mm lateral of bregma. The craniotomy was cleaned with SurgiFoam (Ethicon) soaked in phosphate-buffered solution (PBS). If the dura remained intact, a durotomy was performed.

Virus (AAV9-CaMKII-Cre, diluted to 1×10$^{13}$ particles per mL in PBS, Addgene stock 105558-AAV9) was then pressure injected (NanoJect III, Drummond Scientific) at 2–6 sites near the target site, with 140 nL injected at each of two depths per site (250 and 500 μm below the pia) over 5 min each. The injection pipette was kept in place for another 3–6 min to ensure viral dispersion before being removed slowly over 2–3 min. In mice numbers 3–5 an additional virus injection (AAVretro-tdTomato, diluted to 1×10$^{13}$ particles per mL in PBS, Addgene stock 59462-AAVrg) of 60–100 nL was made targeting the bulk of forelimb somatosensory cortex at –0.73±0.08 mm posterior and ~2.69±0.15 mm lateral of bregma. This injection retrogradely labeled cell bodies that sent projections to S1-fl. In this paper, the labeling was used solely for stabilizing the imaging plane (see below) in both areas. In mice 1 and 2, a small craniotomy was made using a dental drill over the right M1-fl at 0.4 mm anterior and 1.6 mm lateral of bregma. 140 nL of AAVretro-tdTomato (1×10$^{13}$ particles per mL, Addgene) was then injected at 300 μm below the pia. This injection labeled cells in the left M1-fl projecting to the contralateral M1-fl. As above, this labeling was used solely for stabilizing the imaging plane.

The injection craniotomy was sealed with a drop of Kwik-Cast (World Precision Instruments). For mice 1–3 and the two VGAT-ChR2 mice, the cranial window craniotomy was sealed with a custom cylindrical glass plug (3 mm in diameter, 660 µm in depth; Tower Optical) bonded (Norland Optical Adhesive 61, Norland) to a 4 mm number 1 round coverslip (Harvard Apparatus) that was glued to the skull surface with tissue glue. For mice 4 and 5, the craniotomy was sealed with a custom metal and glass cannula created from a small stainless steel ring (3.96 mm outer diameter, 3.45 mm inner diameter, 0.76 mm length, MicroGroup) bonded to a 4 mm number 1 round coverslip using cyano-acrylate glue (ZAP-A-GAP, Robart). All window implants were glued in place first with tissue glue (VetBond) and then with cyanoacrylate glue (Krazy Glue) mixed with dental acrylic powder (Ortho Jet; Lang Dental). In all mice, two layers of MetaBond (Parkell) were applied, and then a custom laser-cut titanium head bar was affixed to the skull with black dental acrylic. Animals were awoken by administering atipamezole via intraperitoneal injection and allowed to recover at least 3 days before water restriction.

## Behavioral task

As previously reported (*Zhu et al., 2022*), the behavioral task (*Figure 1*) was a variant of the water-reaching task of *Galiñanes et al., 2018* that we term the 'water grab' task. This task was performed by water-restricted, head-fixed mice, with the forepaws beginning on paw rests (eyelet screws) and the hindpaws and body supported by a custom, 3D-printed, tapered, clear acrylic tube enclosure. After holding the paw rests for 700–900 ms, a tone was played by stereo speakers and a 2- to 3 µl droplet of water appeared at one of two water spouts (22 gauge, 90 degree bent, 1-in blunt dispensing needles, McMaster) positioned on either side of the snout. The pitch of the tone indicated the location of the water, with a 4 kHz tone indicating left and a 7 kHz tone indicating right. The cue tone lasted 500 ms or until the mouse made contact with the correct water spout, whichever came first. The mouse could grab the water droplet and bring it to its mouth to drink any time after the tone began. Both the paw rests and spouts were wired with capacitive touch sensors (Teensy 3.2, PJRC). Good contact with the correct spout produced an intertrial interval of 3–6 s, while failure to make contact (or insufficiently strong contact) with the spout produced an intertrial interval of 20 s. Because the touch sensors required good contact from the paw, this setup encouraged complex contacts with the spouts. Control software was custom written in MATLAB R2018a using PsychToolbox 3.0.14. Touch event monitoring and task control were performed at 60 Hz using the Teensy.

## Behavioral task training

The mice were trained to make all reaches with the right paw and to keep the left paw on the paw rest during reaching. For the first 1–3 days, mice were head-fixed and provided water directly to their mouths to acclimate them to head-fixation and reward collection. During this period, the animals were encouraged to make consistent contact with both paw rests to trigger water delivery at their mouths. After acclimation and consistent trial initiation, the spout was withdrawn to a position on the right side of their face intended to elicit forelimb reaching with the right arm. If animals were reluctant to reach, small droplets of water were placed on their right whisker pad with a blunt syringe to induce grooming and water collection. After animals reliably performed reaches toward the rightward spout location (>80–90% success), another spout was installed near the initial location (the 'left' spout). Water was then solely delivered through the left spout and its position was shifted across the midline to the ventral contralateral whisker pad over the course of 7–10 days. Once the left spout was in a nearly equidistant location from the midline as the right spout, the animal was cued to reach to both spouts in blocks of 10–50 trials. After reaching >90% in blocked trials, trials were cued randomly. This training procedure took 2–4 weeks, although the behavior continued to solidify for at least two more weeks. Data presented here were collected after 6–8 weeks' experience with the task.

## Optogenetic inhibition

Two VGAT-ChR2 mice were trained to perform the task. After 6–8 weeks of training, inactivations were performed using the epifluorescence path of our 2-photon microscope, centered over coordinates 1.5 L 0.5 A from bregma. The stimulation protocol was designed to approximately replicate the experiments of *Guo et al., 2015*. Pulses were 17 mW peak power, 12.5 ms long, and delivered at 40 Hz (50% duty cycle, 8.5 mW average power). The spot of light was approximately 1.3 mm in diameter at

the focal plane. Stimulation was delivered beginning 150 ms before the cue and then pulsed continuously for 3 s after the cue. After an initial behavioral warm-up period of 50 control trials, stimulation was delivered with 25% probability on subsequent trials. To compensate for visible blue light due to leaks around the light-sealing cone or through the brain, two bright blue LED strips were placed in front of the animal and illuminated constantly throughout the session.

### High-speed videography and stereo triangulation

As previously reported (*Zhu et al., 2022*), high-speed video data was recorded using a pair of cameras (BFS-U316S2M-CS, FLIR; varifocal lenses COZ2813CSIR2, Computar) mounted 150 mm from the right paw rest at 10° apart. Infrared illuminators enabled behavioral imaging while performing 2 p imaging in a darkened microscope enclosure. Cameras were synchronized and recorded at 150 frames per second with real-time image cropping and JPEG compression and streamed to one HDF5 file per camera (areaDetector module of EPICS, CARS). DeepLabCut (DLC) (*Mathis et al., 2018*) was used to track 15 keypoints in each image (see next section). The MATLAB Stereo Camera Calibration toolbox was used for estimation of stereo camera parameters to triangulate paired keypoints into 3D.

### Markerless tracking

3–5 days before recording, the right forelimb of each mouse was shaved and depilated using Nair while the animal was under light anesthesia (1–2% isoflurane). We did not observe any obvious impacts of this depilation on the behavior. DLC was then used to identify 15 key points associated with bony landmarks spanning the proximal and distal forelimb. These key points included: the tip (#1–4), proximal interphalangeal (#5–8) (PIP) and metacarpophalangeal joint (#9–12) (MCP) of each digit, the center of the wrist (#13), distal edge of the elbow joint (#14), and proximal tip of the lesser tubercle of the humerus (#15). We developed a custom GUI in Python to improve the quality of manually labeled training frames. This GUI allowed simultaneous labeling in both images and visualized the epipolar lines of each keypoint in each image to improve stereo correspondence. We then employed an active learning approach to create a dataset of manually labeled images. First, 20–80 frames paired between cameras were randomly selected from all 12 datasets. These images were manually labeled for all 15 markers, and an initial DLC model was trained to identify keypoint locations for both cameras. We then performed three rounds of active learning, selecting 15–35 additional paired frames per dataset per round. We selected frames for inclusion in each round by identifying 3D postures in each video that strongly differed from those already contained in the training set (*Feng et al., 2023*). We assessed the similarity of candidate frames in a window of 300–1000 ms around the water cue to the training set by computing the sum of Euclidean distances between vectorized 3D keypoint positions. We then sorted candidate frames by this distance and manually selected additional frames to label from the 10% of candidate frames with the greatest total distance. At the end of this process, we labeled a total of 1455 paired images across 12 datasets (100–165 images per dataset) for a total of 2910 high-quality labeled images.

### Joint angle kinematics

To extract joint angle kinematics from the 3D triangulated keypoints, we first obtained vectors connecting adjacent keypoints in the unsmoothed data. Because we chose keypoint locations to closely align with putative joint centers across the forelimb, the angles between vectors connecting keypoints were taken to approximate the angles associated with the joint.

For the shoulder joint, we identified the 'humerus' with the vector pointing from the shoulder keypoint to the elbow keypoint. For the elbow joint, we identified the 'radius/ulna' with the vector from elbow to wrist. The vector used to compute wrist angles pointed from wrist to the mean of the second and third MCP keypoints (virtual MC or vMC). For each digit, we computed the vector connecting the corresponding MCP to PIP keypoints, and finally the PIP to DIP keypoints. The vectors defined in this way were used to compute extrinsic Euler angles around each of the putative joint centers in a proximal-to-distal fashion. After each joint angle was computed, the corresponding vector and all of its children vectors were rotated so as to neutralize the rotation around that joint.

We chose a coordinate system appropriate for our task in mice. Starting with the shoulder joint, we computed the abduction/adduction of the humerus as the rotation around the external (world) Y axis (green vector in *Figure 3A*) necessary to bring the humerus into the ZY plane; abduction

was chosen to be negative and adduction was positive. Next, we computed the flexion/extension as the rotation around the X axis (red vector in *Figure 3A*) required to align the humerus with the Z axis (blue vector in *Figure 3A*); flexion was positive. Finally, we computed the shoulder rotation as the angle around the Z axis required to bring the radius/ulna into the YZ plane; internal rotation was positive. At the elbow, we computed flexion/extension as the angle around the X axis required to make the radius/ulna parallel to the Y axis; flexion was positive. We followed a similar procedure for the wrist as the shoulder; abduction/adduction was computed as the angle around the X axis required to bring the vMC into the XY plane; radial deviation or adduction was positive. We computed the wrist flexion/extension as the angle around the Z axis that brought the vMC into the YZ plane; flexion was positive. Finally, to compute the wrist rotation, we first obtained the vector normal to the plane spanned by the second and third MCP vectors (nMCP). Wrist 'rotation' (torsion of the radius around the ulna) was computed as the angle around the Y axis required to bring the nMCP into the YZ plane; pronation was positive. We then computed two intermetacarpal angles, quantifying how 'opposed' MCP1 or MCP4 was to MCP2 or MCP3,, respectively. This measure quantifies the conical deformation of the paw. This was done by computing the angle around the Y axis required to bring the MCP1 or MCP4 vectors into the YZ plane so that they were coplanar with MCP2 and MCP3. We calculated paw splay for the metacarpal 1–2, 2–3, and 3–4 vectors as the angle around the Z axis required to bring them into the XY plane. We then computed two angles around each MCP joint: abduction/adduction and flexion/extension. MCP abduction/adduction was computed as the angle around the X axis required to bring the proximal phalanx into the XY plane; abduction from the wrist midline was positive. We computed the metacarpal flexion/extension as the angle around the Z axis required to bring the proximal phalanx vector into the YZ plane; flexion was positive. Finally, we computed the PIP flexion as the angle between the distal phalanx and the proximal phalanx for each digit.

## Trial screening

To identify trials with problematic keypoint tracking, we used the extracted joint angle kinematics. We first binned joint angles for each trial at 10 ms in a window from –100 to 400 ms locked to lift onset and smoothed with a 15 ms Gaussian kernel. Left and right trials were considered separately. For each joint angle, we found the 2.5 and 97.5 percentile values (pctile1 and pctile2) across all trials and time points. We then computed threshold values for each joint angle as (pctile1+pctile2)/2±abs(pctile1-pctile2). Across all trials, we identified individual joint angle values that exceeded the bounds of these two thresholds. Next, for each trial, we computed the number of time bins for which any joint angle was outside the bounds. If a trial had more than five such time bins, it was excluded from further analysis. Across all mice, this procedure identified 4719/4993 trials with high quality tracking, which were further confirmed by visual inspection. After screening, no substantial tracking errors were detected even on single frames, and thus no further cleaning was performed.

## Joint angle correlations and dimensionality

As in the screening procedure, we first binned at 10 ms, locked joint angles from –100 to 400 ms relative to lift onset, and smoothed each angle within trials with a 15 ms Gaussian kernel. We then concatenated all trials across time. To exclude unusual joint angle configurations due to contact forces, we removed time points during which the paw made contact with the spout target. We then computed the correlation matrix shown in *Figure 3* using the *corrcoef* MATLAB function on this concatenated matrix. To compute the dimensionality of the joint angles and joint velocities, we performed double cross-validated principal component analysis (PCA) on the same concatenated matrix (*Yu et al., 2009*). To perform this procedure, we first held out a subset of observations and performed PCA. Then, for each held-out observation, we held out one joint, used the values of the other joints together with the loadings to estimate the low-D coordinates on that observation via regression, and finally estimated the value of the held-out joint in the held-out observations from the low-D coordinates and the loading for that joint. After cycling over all variables and five random observation partitions for each candidate dimensionality, we identified the optimal dimensionality as the one that minimized the reconstruction error of the held-out variable in the held-out observations. This procedure was performed separately for joint angles and joint velocities.

## Wide-field imaging for area identification

To identify the forelimb primary somatosensory cortex, we performed wide-field imaging while providing sensory input to the contralateral paw using a vibration motor (SparkFun, part ROB-08449) glued to a paw rest (*Alonso et al., 2023*). This provided vibrotactile stimulation of the entire forelimb. The mouse was in the type of behavioral setup they were accustomed to, except for the paw rest, and did not have to perform any task. Vibration was delivered for 250 ms once the mouse's paws had been on the paw rests for at least 2 s.

The widefield macroscope was identical to the setup of *Musall et al., 2019*, except that a pco. panda 4.2 sCMOS camera (Excelitas) was substituted and LED triggers were delivered by a Teensy 3.2. Imaging was performed at 30 frames/s, alternating blue and violet illumination. Violet frames were used for hemodynamic correction.

We used the locus of stimulation-evoked activity to align each cranial window to the Allen CCF; see *Figure 4—figure supplement 1*. The location of strongest activation was aligned with the center of the widest part of the CCF area named SSp-ul (roughly 0 AP 2.6 ML relative to bregma), where proprioceptive and tactile responses most strongly overlap. We targeted our S1 imaging fields of view to this location, as done in *Alonso et al., 2023*, and referred to the recorded region as forelimb S1 or S1-fl. With the window aligned to the atlas, we identified M1-fl with the coordinates of 0.4 AP and 1.6 ML relative to bregma. We targeted our M1-fl imaging field of views to this region. In some cases, thick vasculature or weaker indicator expression prevented high-quality imaging of the ideal location. In these few cases, we imaged as close as possible to the target. Results were similar between all datasets.

## Two-photon imaging

Calcium imaging procedures have been described previously. Briefly, imaging was performed with a Neurolabware 2 p microscope running Scanbox 4.1 and a pulsed Ti:sapphire laser (Vision II, Coherent). Depth stability of the imaging plane was maintained using a custom plugin that made automatic movements of the objective up to every ~10 s based on comparing acquired red-channel data with an initially acquired image stack. Offline, images were run through Suite2p to perform motion correction, region-of-interest (ROI) detection and fluorescence extraction. ROIs were manually curated using the Suite2p GUI. As previously described (*Zhu et al., 2022*), fluorescence was neuropil-corrected, detrended, and passed through OASIS using the 'thresholded' method, AR1 event model, and limiting the tau parameter to be between 300 and 800 ms. Neurons were discarded if they did not meet a minimum SNR criterion. To put events on a more useful scaling, for each ROI, we found the distribution of event sizes, smoothed the distribution (ksdensity in MATLAB, with an Epanechnikov kernel and log transform), found the peak of the smoothed distribution and divided all event sizes by this value. This rescales the peak of the distribution to have a value of unity.

When time-locking the data for visualization, modulation testing, or for decoding analyses, the deconvolved events were resampled into 10 ms bins. This resampling was performed by assigning a fraction of each event into the new 10 ms bins proportionally to how much the 10 ms bins overlapped the original ~32.2 ms frames, and taking into account the exact time each ROI centroid was sampled based on its position within the field of view (FOV). Alignment for RADICaL has been described previously (*Zhu et al., 2022*).

## Modulation index

To test whether regions of interest (ROIs) were task responsive, we adapted the 'ZETA' method from *Montijn et al., 2021* to time series data, closely related to the approach used in *Heimel et al., 2023*. The goal of this procedure was to determine whether the activity of each ROI was reliably modulated over time relative to some task event, such as cue or paw lift. First, neural data for all trials were resampled to 10 ms as described in the previous section. For each ROI, we averaged activity over trials and computed its fractional cumulative sum over time bins. This cumulative sum was compared to the linear baseline that would be expected from an ROI that fired uniformly at a matched rate in the same time window. The deviation of the cumulative sum from the linear baseline (i.e. the residuals) was computed, then this deviation trace was mean-centered and its maximum absolute value was found. This value was the data maximum deviation. We then generated a null distribution for this statistic. To do so, we shuffled by circularly permuting the event-locked data independently for each trial before

averaging across trials, mean-centering, and computing the shuffle's maximum deviation as above. This procedure was repeated 100 times to produce the distribution for the maximum deviation under the null hypothesis that the cell fired uniformly over time. We then computed the p-value as the probability of observing the data maximum deviation under a Gumbel distribution parameterized by the mean and variance of the distribution of maximum deviations over shuffles. Compared to the ZETA procedure in *Heimel et al., 2023*, our procedure may be slightly less sensitive for small time windows because of the disruption of temporal correlation at the circular permutation wraparound point. However, it uses data only from within the time window of interest, so is less sensitive to unrelated modulation due to events occurring just outside the time window of interest, and it is much faster to compute. We performed this procedure for cue-locked (–200 to +400 ms), lift-locked (–200 to +400 ms), and first-contact-locked (–300 to 300 ms) data, for left and right trials separately. P-values obtained through this procedure were then Bonferroni corrected for dual tests when measuring the number of cells modulated to a given event and corrected for six tests (two targets and three events) when measuring the overall number of modulated cells. Note that we did not correct for the number of ROIs tested for two reasons. First, the goal of this testing was to serve as a criterion for inclusion in subsequent decoding analyses, not to determine whether any neurons in the area at all were modulated; and second, correcting for the number of ROIs would bias comparison between areas if different numbers of ROIs were recorded in one area versus the other.

## RADICaL modeling

RADICaL models were fit using the same procedures developed in *Zhu et al., 2022*, except we trained these models using the NeuroCAAS implementation (*Abe et al., 2022*). As in *Zhu et al., 2022*, we used lift-locked data binned at 10 ms in a window from –100 to +800 ms around lift time with 100 ms padding. After models were fit, the padding was removed and RADICaL-inferred rates for each ROI were handled identically to sub-frame aligned smoothed and deconvolved data. Note that all results were qualitatively similar without RADICaL, but as expected, decoding quality and correlations were generally somewhat lower.

## Trial grouping

We observed considerable heterogeneity in the time from lift to spout contact (movement time) for both conditions for all mice. We reasoned that this could follow from single-trial heterogeneity in either reach shape or accuracy. We, therefore, separated trials within each condition into two groups, determined by whether their movement time was smaller or greater than the median movement time for that condition. This procedure yielded four groups of trials, 'brief' and 'long' for left and right conditions.

## Peri-event time histograms

PETHs were computed for three different locking events: cue, lift, and first contact. Neural data for all trials were binned at 10 ms and smoothed with a Gaussian (35 ms s.d.). Cue and lift-locked PETHs were locked in a window of –100–700 ms relative to the corresponding event. Contact-locked PETHs were locked in a window of –400–400 ms relative to first contact. Trials were then averaged within the groups described above.

## Joint angle and velocity decoding

Joint angle or velocity kinematics were linearly interpolated from their original 6.66–10 ms and smoothed with a Gaussian (15 ms s.d.). These angular variables were then treated linearly in decoding analyses as their ranges were relatively constrained during the reaching and grasping movements; although the true relationships are likely nonlinear, this serves as a sufficient approximation to demonstrate the presence of a relationship between neural activity and kinematics. We then screened for modulated neurons. ZETA p-values were sorted ascending, and the top 100 cells were included in regression models in order to use equal numbers of neurons across areas. For decoding, neural and kinematic data were locked to –100 ms before and 400 ms lift onset. Deconvolved data were smoothed with a Gaussian (35 ms s.d.); RADICaL-inferred rates were not smoothed. We did not include a lag between neural activity and kinematics for simplicity; tests on selected sessions indicated that adding lags did not improve decoding. We applied PCA to the neural activity and reduced its dimensionality

to be 90% of the number of included neurons. The decoder was trained and tested using cross-validated ridge regression. We partitioned the trials into 10 different non-overlapping training (90%) and test (10%) set splits and fit the model weights and regularization parameter on the training set using a heuristic described in *Karabatsos, 2018* with code developed in *Musall et al., 2019*. We then applied the trained decoder to the test set and computed the fraction variance explained for each joint angle or velocity independently. We then repeated this same procedure for four additional 10-fold partitions for a total of 50 cross-validated folds. We report the variance explained across joints for all folds. To visualize reconstructions of the joint angles using the fit decoders, for each trial, we 'bagged' the decoder weights: we averaged across the five cross-validation folds where that trial was in the test set. This produced a denoised decoder that was used to predict the joint kinematics for that trial.

## Target decoding analysis

We used logistic regression to classify individual time points as belonging to rightward or leftward targets from population activity. Neurons with significant modulation at a ZETA $p<0.01$ threshold for any locking event and either target were included. Neural data from each trial were binned at 10 ms, locked to a window of −100–500 ms centered on cue onset, and smoothed with a Gaussian (35 ms s.d.). We partitioned the trials into five non-overlapping training (80%) and test (20%) set splits, stratifying left and right trials. We then trained a logistic regression classifier using the MATLAB *fitclinear* function on all post-cue time points across training set trials. We then projected individual trials onto the classifier dimension. We normalized each trial's projection according to the 90th percentile value over all time points of the same trial type. We report the classifier performance on all time points across test set trials. Our procedure for computing the threshold crossing times for individual trials was similar to the one used in *Kaufman et al., 2015*. We computed the threshold value by first taking the median of all projections across trials at each time point, finding the maximum and minimum values of this trace, then computing their midpoint; this procedure was done separately for left and right trials. We found the threshold crossing time as the first time bin after which the projection stayed above the threshold for at least five consecutive bins (50 ms). If trials never crossed the threshold or did not stay above the threshold for more than five bins, they were discarded. We then interpolated between 10 ms bins to find the exact time that the projection crossed the threshold.

## Time-generalized target decoding analysis

To assess the stability of the target-classifier dimension over time, we employed a similar logistic regression approach as above. Neural data from each trial were first interpolated as usual to 10 ms bins from −100–1000 ms relative to the cue, then smoothed with a Gaussian (35 ms s.d.). Time bins were then grouped into 11 non-overlapping 100 ms epochs. For each epoch, logistic regression was used to classify individual 10 ms time bins as belonging to left or right trials. A separate classifier was trained for each epoch, and thus 11 models were fit in total. This procedure was performed using fivefold cross-validation, with left and right trials stratified. The trained models for each epoch were used to predict the target identity on time points across all epochs for the held-out 20% of trials from each fold. We reported the mean performance of the classifier over folds for each prediction epoch.

## Reaction time decoding analysis

Deconvolved neural data were used to predict the lift reaction time (RT) on single trials. Our approach follows closely the procedure used in *Zhu et al., 2022* based on *Kaufman et al., 2016*. A cross-validated logistic regression classifier was trained to discriminate pre-lift neural data (a bin −100–0 ms relative to lift onset) from post-lift neural data (a bin 0–100 ms relative to lift onset). Training was performed with *fitclinear* in Matlab, stratifying left and right trials. Neural data from the held-out trials was interpolated at 10 ms as usual, locked −200–600 ms relative to the lift, smoothed with a Gaussian (10 ms s.d.), then projected onto this dimension. The lift RT was calculated as the time that the projection crossed a threshold. To compute the threshold, we calculated the median across trials, then took the midpoint of the maximum and minimum. We summed the predicted time (which was effectively an offset from the true RT) with the ground truth lift RTs and correlated these with the ground truth lift RTs.

## Acknowledgements

We thank M Rivers and R Vescovi for assistance with installing and configuring the high-speed camera platform, D Sabatini for help setting up the behaviors and discussions of analysis, P Ravishankar for animal care assistance, S Musall for widefield imaging code, and the Maunsell, MacLean, Sheffield, and Giovannucci Labs for aid with reagents, transgenics, image debugging, and calcium data preprocessing. This work was funded by The University of Chicago, NSF-NCS 1835390 (MK), NIH-NINDS R01 NS121535 (MK), the Simons Foundation (MK), the Sloan Foundation (MK), the Whitehall Foundation (MK), the NSF-Simons National Institute for Theory and Mathematics in Biology via grants NSF DMS-2235451 and Simons Foundation MP-TMPS-00005320 (MK), and NIH T32 NS121763 (HG).

## Additional information

### Funding

| Funder | Grant reference number | Author |
|---|---|---|
| National Science Foundation | NCS 1835390 | Matthew T Kaufman |
| National Institute of Neurological Disorders and Stroke | R01 NS121535 | Matthew T Kaufman |
| Simons Foundation | 876393SPI | Matthew T Kaufman |
| Alfred P. Sloan Foundation | | Matthew T Kaufman |
| Whitehall Foundation | | Matthew T Kaufman |
| National Science Foundation | DMS-2235451 | Matthew T Kaufman |
| Simons Foundation | MP-TMPS-00005320 | Matthew T Kaufman |
| National Institute of Neurological Disorders and Stroke | T32 NS121763 | Harrison A Grier |

The funders had no role in study design, data collection and interpretation, or the decision to submit the work for publication.

### Author contributions

Harrison A Grier, Data curation, Software, Formal analysis, Validation, Investigation, Visualization, Methodology, Writing – original draft, Writing – review and editing; Sohrab Salimian, Data curation, Software, Formal analysis, Visualization; Matthew T Kaufman, Conceptualization, Resources, Data curation, Software, Supervision, Funding acquisition, Investigation, Visualization, Methodology, Writing – original draft, Project administration, Writing – review and editing

### Author ORCIDs

Harrison A Grier ⓘ https://orcid.org/0000-0001-9107-2912
Sohrab Salimian ⓘ https://orcid.org/0000-0002-9685-0581
Matthew T Kaufman ⓘ https://orcid.org/0000-0002-8072-023X

### Ethics

All procedures were approved by the University of Chicago Institutional Animal Care and Use Committee (Animal Care and Use Protocol #72555).

Reviewer #1 (Public review): https://doi.org/10.7554/eLife.106270.3.sa1
Reviewer #2 (Public review): https://doi.org/10.7554/eLife.106270.3.sa2
Author response https://doi.org/10.7554/eLife.106270.3.sa3

## Additional files

**Supplementary files**
MDAR checklist

**Data availability**
Data are available on figshare. Code used to create the figures is available on GitHub: (https://github.com/kaufmanlab/GSK25-public copy archived at *Grier, 2025*).

The following dataset was generated:

| Author(s) | Year | Dataset title | Dataset URL | Database and Identifier |
|---|---|---|---|---|
| Grier HA, Salimian S, Kaufman MT | 2025 | Data from "Mouse sensorimotor cortex reflects complex kinematic details during reaching and grasping" eLife 2025 | https://doi.org/10.6084/m9.figshare.28326998 | figshare, 10.6084/m9.figshare.28326998 |

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
