## [Editor Report · eLife Assessment]

The granularity with which neural activity in the sensorimotor cortex of mice corresponds to voluntary forelimb motion is a key open question. This paper provides **compelling** evidence for the encoding of low-level features like joint angles and represents an **important** step forward toward understanding cortical limb control signals.

---

## [Referee Report · Reviewer #1 (Public review)]

Summary:

This study addresses the encoding of forelimb movement parameters using a reach-to-grasp task in mice. The authors use a modified version of the water-reaching paradigm developed by Galinanes and Huber. Two-photon calcium imaging was then performed with GCaMP6f to measure activity across both the contralateral caudal forelimb area (CFA) and the forelimb portion of primary somatosensory cortex (fS1) as mice perform the reaching behavior. Established methods were used to extract the activity of imaged neurons in layer 2/3, including methods for deconvolving the calcium indicator's response function from fluorescence time series. Video-based limb tracking was performed to track the positions of several sites on the forelimb during reaching and extract numerous low-level (joint angle) and high-level (reach direction) parameters. The authors find substantial encoding of parameters for both the proximal and distal parts of the limb across both CFA and fS1, with individual neurons showing heterogeneous parameter encoding. Limb movement can be decoded similarly well from both CFA and fS1, though CFA activity enables decoding of reach direction earlier and for a more extended duration than fS1 activity. Collectively, these results indicate involvement of a broadly distributed sensorimotor region in mouse cortex in determining low-level features of limb movement during reach-to-grasp.

Strengths:

The technical approach is of very high quality. In particular, the decoding methods are well designed and rigorous. The use of partial correlations to distinguish correlation between cortical activity and either proximal or distal limb parameters or either low- or high-level movement parameters was very nice. The limb tracking was also of extremely high quality, and critical here to revealing the richness of distal limb movement during task performance.

The task itself also reflects an important extension of the original work by Galinanes and Huber. The demonstration of a clear, trackable grasp component in a paradigm where mice will perform hundreds of trials per day expands the experimental opportunities for the field. This is an exciting development.

The findings here are important and the support for them is solid. The work represents an important step forward toward understanding the cortical origins of limb control signals. One can imagine numerous extensions of this work to address basic questions that have not been reachable in other model systems.

Collectively, these strengths made this manuscript a pleasure to read and review.

---

## [Referee Report · Reviewer #2 (Public review)]

Summary:

In this manuscript, Grier, Salimian, and Kaufman characterize the relationship between the activity of neurons in sensorimotor cortex and forelimb kinematics in mice performing a reach-to-grasp task. First, they train animals to reach to two cued targets to retrieve water reward, measure limb motion with high resolution, and characterize the stereotyped kinematics of the shoulder, elbow, wrist, and digits. Next, they find that inactivation of the caudal forelimb motor area severely impairs coordination of the limb and prevents successful performance of the task. They then use calcium imaging to measure the activity of neurons in motor and somatosensory cortex, and demonstrate that fine details of limb kinematics can be decoded with high fidelity from this activity. Finally, they show reach direction (left vs right target) can be decoded earlier in the trial from motor than from somatosensory cortex.

Strengths:

In my opinion, this manuscript is technically outstanding and really sets a new bar for motor systems neurophysiology in the mouse. The writing and figures are clear, and the claims are supported by the data. This study is timely, as there has been a recent trend towards recording large numbers of neurons across the brain in relatively uncontrolled tasks and inferring a widespread but coarse encoding of high-level task variables. The central finding here, that sensorimotor cortical activity reflects fine details of forelimb movement, argues against the resurgent idea of cortical equipotentiality, and in favor of a high degree of specificity in the responses of individual neurons and of the specialization of cortical areas.

Comment on revised version:

The authors addressed all my concerns, and in my opinion, the manuscript is suitable for publication of the Version of Record in its current form.

---

## [Author Response]

The following is the authors’ response to the original reviews.

**Public Reviews:**

**Reviewer #1 (Public review):**
Summary:This study addresses the encoding of forelimb movement parameters using a reach-to-grasp task in mice. The authors use a modified version of the water-reaching paradigm developed by Galinanes and Huber. Two-photon calcium imaging was then performed with GCaMP6f to measure activity across both the contralateral caudal forelimb area (CFA) and the forelimb portion of primary somatosensory cortex (fS1) as mice perform the reaching behavior. Established methods were used to extract the activity of imaged neurons in layer 2/3, including methods for deconvolving the calcium indicator's response function from fluorescence time series. Video-based limb tracking was performed to track the positions of several sites on the forelimb during reaching and extract numerous low-level (joint angle) and high-level (reach direction) parameters. The authors find substantial encoding of parameters for both the proximal and distal parts of the limb across both CFA and fS1, with individual neurons showing heterogeneous parameter encoding. Limb movement can be decoded similarly well from both CFA and fS1, though CFA activity enables decoding of reach direction earlier and for a more extended duration than fS1 activity. Collectively, these results indicate involvement of a broadly distributed sensorimotor region in mouse cortex in determining low-level features of limb movement during reach-to-grasp.Strengths:The technical approach is of very high quality. In particular, the decoding methods are well designed and rigorous. The use of partial correlations to distinguish correlation between cortical activity and either proximal or distal limb parameters or either low- or high-level movement parameters was very nice. The limb tracking was also of extremely high quality, and critical here to revealing the richness of distal limb movement during task performance.The task itself also reflects an important extension of the original work by Galinanes and Huber. The demonstration of a clear, trackable grasp component in a paradigm where mice will perform hundreds of trials per day expands the experimental opportunities for the field. This is an exciting development.The findings here are important and the support for them is solid. The work represents an important step forward toward understanding the cortical origins of limb control signals. One can imagine numerous extensions of this work to address basic questions that have not been reachable in other model systems.Collectively, these strengths made this manuscript a pleasure to read and review.

Thank you!

Weaknesses:In the last section of the results, the authors purport to examine the representation of "higher-level target-related signals," using the decoding of reach direction. While I think the authors are careful in their phrasing here, I think they should be more explicit about what these signals could be reflecting. The "signals" here that are used to decode direction could relate to anything - low-level signals related to limb or postural muscles, or true high-level commands that dictate only what movement downstream motor centers should execute, rather than the muscle commands that dictate how. One could imagine using a partial correlation-type approach again here to extract a signal uncorrelated with all the measured low-level parameters, but there would still be all the unmeasured ones. Again, I think it is still ok to call these "high-level signals," but I think some explicit discussion of what these signals could reflect is necessary.

Thank you for this excellent suggestion. We have followed both pieces of the reviewer’s advice. First, we performed the suggested analysis, partialing off the kinematics then performing target classification on the residuals. This is now Figure 6S1. The analysis revealed the presence of target-related information in the neural activity after subtracting off all linear correlations with kinematics, supporting our claims that higher-level information is present in both populations. The exact timing of classifier performances varied substantially across mice, potentially due to differences in reach-to-grasp strategy, kinematic tracking fidelity, and exact spatial locations of each recorded FOV. Following the second suggestion, we have made the relevant text more careful. We now conclude simply that higher-level signals, meaning those signals that are largely unrelated to forelimb joint angle kinematics, are present but with variable timing and strengths in each area. That text now reads:

“Target decoding performance could result from truly higher-level signals that code abstractly for target location, or alternatively could be supported by strong encoding of kinematic variables that differed between targets. To disambiguate these possibilities, we refit the linear classifier to neural data after regressing off variance related to the joint angle kinematics. The strength and exact time course of the resulting target decoding varied somewhat across animals, but the earliest portion of target decoding performance persisted in all animals after the removal of kinematics and performance remained stronger for M1-fl than S1-fl (Figure 6-figure supplement 1B). We thus conclude that higher-level signals are present in both areas, but differ in their exact timing and strength. However, we note that other possible signals, such as postural changes, could not be controlled for here.”

Related to this, I think the manuscript in general does not do an adequate job of explicitly raising the important caveats in interpreting parametric correlations in motor system signals, like those raised by Todorov, 2000. The authors do an expert job of handling the correlations, using PCA to extract uncorrelated components and using the partial correlation approach. However, more clarity about the range of possible signal types the recorded activity could reflect seems necessary.

This is an important point, and our text could have unintentionally misled readers. We have now attempted to make this point explicit in the Discussion and in the Results for Figure 6. This Discussion text now reads:

“Moreover, as is widely known (Todorov 2000), the exact role of these kinematically-related signals is challenging to determine from correlative measures alone; thus, determining whether these signals are used for direct movement control or instead indirectly reflect control performed elsewhere is left as a topic for future work.”

The manuscript could also do a better job of clarifying relevant similarities and differences between the rodent and primate systems, especially given the claims about the rodent being a "first-class" system for examining the cellular and circuit basis of motor control, which I certainly agree with. Interspecies similarities and differences could be better addressed both in the Introduction, where results from both rodents and primates are intermixed (second paragraph), and in the Discussion, where more clarity on how results here agree and disagree with those from primates would be helpful. For example, the ratio of corticospinal projections targeting sensory and motor divisions of the spinal cord differs substantially between rodents and primates. As another example, the relatively high physical proximity between the typical neurons in mouse M1 and S1 compared to primates seems likely to yoke their activity together to a greater extent. There is also the relatively large extent of fS1 from which forelimb movements can be elicited through intracortical microstimulation at current levels similar to those for evoking movement from M1. All of these seem relevant in the context of findings that activity in mouse M1 and S1 are similar.

We understand two points to address here. The first point is that we needed to be more careful to attribute previous results as being from the rodent vs. monkey. We agree. We have now revised several parts of the paper to make these distinctions clearer. The second point is about the potential benefit of a thorough review of the many ways in which primate and rodent sensorimotor systems differ. We entirely agree that this could be useful for the field. However, this is a sizable endeavor and doing it full justice is beyond what we know how to fit in the space allotted for framing our results here. We therefore sought a compromise, acknowledging how our results correspond to existing results in the primate without exhaustively accounting for how they differ. Future work will be necessary to more carefully disambiguate whether species-specific differences are due to biomechanical, neurological, ethological, or as-of-yet undetermined sources. We have incorporated your final specific points about what could produce similar information in M1 and S1 into the Discussion.

“This may simply be a consequence of widely distributed representations of movement across mouse cortex (Musall et al. 2019; Steinmetz et al. 2019; Stringer et al. 2019), including forelimb somatosensory areas, or may be a consequence of the close physical proximity of M1-fl and S1-fl hindering development of functionally distinct representations (Tennant et al. 2011).”

In addition, there are a number of other issues related to the interpretation of findings here that are not adequately addressed. These are described in the Recommendations for improvement.
**Reviewer #2 (Public review):**
Summary:In this manuscript, Grier, Salimian, and Kaufman characterize the relationship between the activity of neurons in sensorimotor cortex and forelimb kinematics in mice performing a reach-to-grasp task. First, they train animals to reach to two cued targets to retrieve water reward, measure limb motion with high resolution, and characterize the stereotyped kinematics of the shoulder, elbow, wrist, and digits. Next, they find that inactivation of the caudal forelimb motor area severely impairs coordination of the limb and prevents successful performance of the task. They then use calcium imaging to measure the activity of neurons in motor and somatosensory cortex, and demonstrate that fine details of limb kinematics can be decoded with high fidelity from this activity. Finally, they show reach direction (left vs right target) can be decoded earlier in the trial from motor than from somatosensory cortex.Strengths:In my opinion, this manuscript is technically outstanding and really sets a new bar for motor systems neurophysiology in the mouse. The writing and figures are clear, and the claims are supported by the data. This study is timely, as there has been a recent trend towards recording large numbers of neurons across the brain in relatively uncontrolled tasks and inferring a widespread but coarse encoding of high-level task variables. The central finding here, that sensorimotor cortical activity reflects fine details of forelimb movement, argues against the resurgent idea of cortical equipotentiality, and in favor of a high degree of specificity in the responses of individual neurons and of the specialization of cortical areas.

Thank you!

Weaknesses:It would be helpful for the authors to be more explicit about which models of mouse cortical function their results support or rule out, and how their findings break new conceptual ground.

We appreciate this feedback and have attempted to make these details clearer through changes to the Introduction and Discussion. One key change is noted below:

“The presence of detailed kinematic signals in the sensorimotor cortex supports a model of mouse sensorimotor cortex in which M1-fl and S1-fl play a strong role in shaping the fine details of reaching and grasping movements.”

**Recommendations for the authors:**

**Reviewer #1 (Recommendations for the authors):**
In addition to the weaknesses noted above, I suggest the authors also address the following:The last results section is generally lacking in statistical support for claims. Statistical support should be added.

Thank you for pointing this out, we have added more statistical support to this section.

The consideration in the Discussion of relevant previous findings and potential explanations for the distal limb signals in mouse sensorimotor cortex is somewhat lacking. There are several specific issues:(1) In contrast to the present study, the studies cited in regards to a lack of motor cortical involvement did not involve dexterous movements - in fact, Kawai et al. explicitly engineered a task that did not involve dexterity to distinguish the role of motor cortex in learning from its known role in dextrous movement execution. In Kawai et al., the authors note one rat who adopted a more dexterous approach to the lever pressing task; in this rat, a motor cortical lesion did cause a longer-lasting reduction in task performance. In additional experiments reported in Kawai's PhD thesis, performance of a dextrous task does erode with motor cortex lesion, as seen in other studies, like the early rodent reaching work of Whishaw and colleagues.(2) Other possible explanations for the persistence of non-dexterous tasks following motor cortical removal are compensation by, or redundant functionality in, other motor system regions.(3) It is also worth noting that stimulation in different regions of mouse M1 and S1 evokes alternately, digit, wrist, and elbow movements in fairly similar proportions (Tennant, 2011), suggesting that descending pathways substantially target spinal circuits that control all forelimb joints.(4) It also seems relevant that although the recovery time course is longer, nonhuman primates also retain substantial hand control after motor cortical removal (e.g. Lashley, 1925; Glees and Cole, 1950; Passingham et al., 1983). Humans of course, appear to be a different story.

These are good points. We have tried to make the Discussion better reflect the tension in the literature, including with this new text:

“However, several other previous results have indirectly suggested that M1 and S1 may be involved in the details of forelimb movement. Performance suffers with inactivation or lesioning of M1 and S1 in skilled, complex manual behaviors (Guo et al 2015, Mizes et al 2024, Whishaw et al 1990) or idiosyncratic use of digits to accomplish non-dexterous tasks (Kawai 2014). The sparing of non-dexterous tasks with these lesions may also reflect redundancy in control as opposed to irrelevance of M1 and S1. Nevertheless, our finding of low-level kinematic information in sensorimotor cortex supports a role for cortex beyond simply providing redundant high-level commands to these subcortical areas.”

We have avoided mentioning points 3 and 4 in the paper; the stimulation results might follow from activating projections not normally involved in this behavior, and discussing primates in this context would require a long list of caveats. We agree that these points are worth thinking about, but are concerned that they are too circumstantial to include in interpreting the results formally.

Although similar decoding performance is achieved using neurons from both CFA and fS1, I am left wondering whether you would do substantially better with CFA using activity at additional preceding time points, or when using exclusively time points from the past. The primary model used here appears to use neural signals from corresponding time points to decode limb parameters, but results seemingly could be different when using preceding time points as regressors.

We appreciate this suggestion and have added the analysis to an additional supplementary panel for Figure 5 (Figure 5-figure supplement 3). Incorporating lags into the decoder via a Wiener filter does indeed improve the decoding performance, but this could simply be due to the increase in the number of predictor variables. This analysis did not, however, further disambiguate M1-fl and S1-fl: the performance improvement was similar across areas for both causal and acausal lag configurations. This could be a consequence of the time resolution of calcium imaging, so further experiments with electrophysiology would be required to rule this possibility out. We now note this new result:

“Including additional causal (-100 ms preceding) and/or acausal (-100 ms preceding to 100 following) lags improved decoding performance modestly and similarly for both areas (Figure 5-figure supplement 3E-F).”

Related to this, I am also worried about the bleeding of signals across time here. If you deconvolve and interpolate between time points, the interpolation seemingly will pull information into the past, up to half the sampling period, which here is on the order of how long it takes signals to travel to and from the limb. The authors do not make any inappropriate claims about the neural signals here reflecting causes or consequences of what is happening at the limb, but readers (like me) will still try to draw these sorts of conclusions. Is it possible that, although decoding from instantaneous signals is similar for the two regions, the M1 signals are actually motor signals related to future limb state while the S1 signals are sensory consequences? Even if many of the relevant details related to conduction times are not known, perhaps the authors could clarify what can and can't be said related to causal interpretation here.

Thank you for suggesting further explanation here. We agree that our interpretation could be made more specific. We have added text in the Discussion section to speak more directly to what can and cannot be concluded from our analyses. In short, it is hard to be certain of lags in calcium imaging data for many reasons, and using recording methods with finer temporal resolution (like electrophysiology) will be necessary for determining the precise temporal relationships between kinematics and neural activity. In the absence of these recordings, we limit our claim to kinematic information being present in M1-fl and S1-fl neural activity and leave determining the causal role of this information to future work.

New clarifying text in the Discussion:

“The use of calcium imaging further prevents strong conclusions about whether activity reflects future limb states or sensory consequences. Confirming this limitation, inclusion of lagged data in the decoding models, whether causal or acausal, resulted in similar performance changes in both areas.”

An alternative reason why lift onset is less decodable in CFA is that CFA activates substantially before lift onset, as has been observed in previous rodent studies (Kargo and Nitz, 2004; Miri et al., 2017; Veuthey et al., 2020), perhaps as some sort of movement preparation. S1, on the other hand, may not have this early activity, and so may show a clearer transient at onset when the hand and limb start to move. This seems more likely than the explanations provided by the authors.

This is a valid possible alternative explanation and we have updated the Discussion to reflect this. This difference in the structure of M1-fl activity versus S1-fl is apparent in the projections of Figure 6A, which show M1-fl projections more clearly aligned to cue-onset than S1-fl projections.

“Our lift time decoding results are consistent with this view and align with recent observations characterizing mouse proprioceptive forelimb cortex, (Alonso et al 2023), although an alternative explanation may be simply that M1-fl activates earlier than S1-fl during reaching (Kargo and Nitz 2004; Miri et al 2017; Veuthey et al 2020).”

To better clarify relevant similarities and differences between the rodent and primate systems, the Introduction could include some of these similarities and differences exposed by the literature currently cited, and the Discussion could include an additional paragraph specifically relating findings here to previous observations in the primate.

We appreciate the reviewer’s thoughtfulness on possible framings of our results. When writing this paper, framing was a major challenge for us and we drafted quite a few versions of the Introduction including some that focused more on mouse-primate comparison. In the end, we decided the most critical function of the Intro was to set up our central question, of “levels-of-sensorimotor-control”. The rich primate literature was valuable here, but getting into a protracted compare-and-contrast exercise quickly became a distraction from the point. Further, we sought to highlight the relevance and importance of the question answered in our work as the mouse has gained prominence for filling gaps that are challenging to address with primates. This paper serves as one of many early steps towards the ultimate goal of revealing general properties of sensorimotor cortical function with the mouse model. We have made some subtle changes to the Introduction that we hope will more clearly communicate this narrative.

We agree that a Discussion paragraph directly relating our results to those in primates would benefit our conclusions and have added one:

“These results expand our understanding of the rodent sensorimotor system and highlight similarities to nonhuman primates. We show here evidence in mice of detailed joint angle kinematic signals from the full forelimb in M1 and S1, as has been shown in macaque cortex during tasks involving reaching and grasping objects (Vargas-Irwin et al. 2010; Saleh et al. 2010, 2012; Goodman et al. 2019; Okorokova et al. 2020). Additionally, the earlier onset of movement-related activity in M1-fl compared to S1-fl is similar to macaque M1 and S1 (Tanji and Evarts 1976). Taken together these results suggest that the mouse can be employed to address questions traditionally explored in primates about how cortical activity encodes detailed movement commands.”

Although this is outside the scope of the present study, it would be interesting to image descending projection neurons to see what signals are conveyed downstream, and to what targets. Some signals observed in layer 2/3 may not be strongly reflected in descending projections.

We agree that recording from descending projection neurons in this task would be of deep interest – and also agree that these experiments are beyond the scope of the present study. We look forward to performing these additional experiments in future work.

Minor:(1) The use of "CFA" and “fS1” is a bit confusing. S1, like M1, is defined primarily based on histological criteria, while CFA is defined by intracortical microstimulation. CFA contains a substantial fraction of fS1, seemingly most of it based on the maps shown in Tennant et al., 2011. This is not really a criticism, as the field has not reached any sort of consensus on this nomenclature yet.

We are similarly unhappy with the inconsistency of the terminology in the field, and struggled with how not to make it worse. After much debate and consultation with colleagues, we decided to use “M1” and “S1” to evoke the century of literature on these areas; and “-fl” to indicate forelimb because it is more intuitive than “-ul” and avoids using the illegible “-ll” for hindlimb (relevant to our subsequent paper). For what we called M1-fl, we recorded where we did because anecdotally we saw similar responses across that swath; but note that this definition is also consistent with the definition of “MOp-ul” found with multimodal mapping by

Munoz-Castaneda (2021), which extends a little anteriorly of MOp as defined by the Allen CCF. As the field continues to mature, we hope future work can converge on a set of shared terms.

(2) Page 4: "Inactivations and lesions of M1 and S1 have shown that M1 is required for the execution of dexterous reach-to-grasp movements" - to me, earlier work from Whishaw and colleagues deserves to be cited here.

We appreciate the suggestion and have updated the references in this section to better reflect the prior work from Whishaw and other researchers.

(3) Page 5: "evoking sufficient trial-to-trial variability to avoid model overfitting." - what I think the authors are referring to here is a particular kind of "overfitting," the consequence of not exploring the full movement space, as opposed to model overfitting from issues with the model-fitting method itself. Rather than just saying overfitting, the authors could be clearer about what they are referring to.

The reviewer is right; the phenomenon we intended to refer to is not properly termed overfitting. Specifically, we meant that data with restricted range does not necessarily express global structure, and models can therefore incorrectly fit them. For example, fitting a linear model to data including many periods of a sine wave will correctly show a zero-slope linear component, but fitting to only a portion of a single cycle will typically yield a nonzero slope. This is not overfitting, is not exactly underfitting (because the relevant structure is barely present in the data, as opposed to missed by an insufficiently powerful model), is not bias (the data are fit well), and is not even necessarily a problem (the local relationship may be what you are interested in). Yet, it does not reflect the larger structure of the data.

We do not know of a standard term for this phenomenon, so instead of dragging the reader through this tangential argument, we have tried to offer a simpler motivation for using multiple targets:

“Assessing the relationship between neural activity and the details of movement requires striking a balance between achieving repeatable behavior and evoking sufficient trial-to-trial variability to broadly sample movement space”.

(4) Page 5: Caudal Forelimb Area should not be capitalized.

Obviated with the change in area nomenclature.

(5) Page 7: "of linearly independent degrees of freedom" - for a neuroscience audience, I think it is better to explicitly mention that the resulting PCs are uncorrelated.

We agree that this section could benefit from clarification. We have attempted to provide additional nuance to indicate what the analysis was intended to test.

“Despite the strong coupling between the proximal and distal joint angles, rich variation remained in the action of different joints over time. The presence of strong correlations across joints suggested that the kinematics may be well described by a smaller number of independent degrees of freedom than the total number of recorded angles. To assess the number of linearly independent (uncorrelated) degrees of freedom amongst the 24 joint angles and velocities, we used double-cross-validated PCA Yu et al. 2009; Methods; Figure 3D, finding intermediate dimensionalities of 7 (median for joint angles) and 10 (velocities; Figure 3E). This is consistent with the idea that joint angles across the limb are coordinated instead of controlled independently, and that this coordination is flexible enough over time to enable accurately performing reaching and grasping to different targets.”

(6) Page 7: In the Results, the authors should mention what indicator is being used, the imaging frame rate, and summarize briefly how cells were defined.

Thank you for the suggestion, these details have been added to the relevant results section for clarity.

“To do so, we recorded neural activity from neurons in layer 2/3 M1-fl extending into the immediately adjacent secondary motor cortex (M2), and the forelimb region of S1 (S1-fl) using two-photon calcium imaging of GCaMP6f-expressing neurons in layer 2/3 (185-230 μm deep, imaged at 31 Hz, cells extracted with Suite2p (Pachitariu et al 2017)).”

(7) Page 7: "corrected at n=2" - n doesn't typically refer to the number of tests, so for clarity I would say "corrected for dual tests."

Thank you for pointing this out, we have corrected the text and added additional explanation in the methods for our approach to determining statistical significance across the targets and locking events.

“P-values obtained through the ZETA were then Bonferroni corrected for dual tests when measuring the number of cells modulated to a given event and corrected for six tests (2 targets and 3 events) when measuring the overall number of modulated cells.”

(8) Page 7: In the Results, when the decoding is introduced, it would be helpful to have a few details without having to hunt through the Methods. For example, were things regularized, how was cross-validation handled, etc?

Thank you for the suggestion, these details have been added to the relevant results section for clarity.

A simple linear regression model related the single-trial joint angles at all time points to single-trial neural activity at the corresponding moments. The model was fit with ridge regression, the ridge penalty was determined via a heuristic (Karabatsos 2018), and performance was measured on held-out trials (80/20 train/test split, 50 folds).

(9) Page 8: I think it is worth noting how much mouse reaching involves shoulder rotation as opposed to movement in other joints, as this seems very different from primates.

Thank you for pointing this out. We think this is mostly a task difference: our mice were in a quadrupedal stance, whereas monkeys are typically asked to reach from a sitting position. We now mention this in the Results.

“Reaching evoked particularly large rotation of the shoulder, likely because the mice reached from a quadrupedal position to targets on either side of the snout.”

(10) Page 8: Should provide quantification to clarify what is meant by "closely tracked."

We have updated the text to indicate that this claim was meant to be qualitative, and to more clearly highlight that the interest here is the first demonstration of the ability to reconstruct valid forelimb postures from decoded joint angles in the mouse. Quantifying the reconstruction properly would require substantially more manual data labeling, and the successful decoding itself demonstrates indirectly that the reconstructions are good enough to obtain the results of interest.

Additionally, we reconstructed the skeletal representation of the forelimb from the decoded joint angles and found that, as intended, the reconstructed postures had strong qualitative resemblance to the true postures, even of “minor” angles like cylindrical paw deformation or digit splay (Figure 5C,G).

(11) Page 8: "Overall, these results suggest that instantaneous movement-related signals are similarly distributed across CFA and fS1." - I know we are being succinct here, but this sentence sounds like a non sequitur in the context of this paragraph - perhaps include a conclusion from the results in this paragraph first, then summarize the whole section.

Thank you for the suggestion, we have updated this text to more clearly conclude the results of this section.

Overall, these results reveal that neural activity in M1-fl and S1-fl is closely related to the kinematic details of reach-to-grasp movements. The ability to decode substantial variance in proximal and distal joints suggests that this relationship extends to the entire forelimb and the similar performance obtained from each area suggests that this information is similarly distributed across M1-fl and S1-fl.

(12) Page 10: Mention of projections from fS1 does not explicitly specify their preferential targeting of the dorsal horn, which seems relevant.

We appreciate the suggestion and have added this detail to the text.

Rodent S1-fl is known to influence interneuron populations in the spinal cord through direct and indirect projections that predominantly target the dorsal horn (Ueno et al. 2018), thus these signals may also reflect S1-fl’s important role in modulating reflex circuits to coordinate sensory feedback with movement generation (Moreno-López et al. 2016; Moreno-Lopez et al. 2021; Seki et al. 2003).

(13) Page 31: Labels on the figure indicating what blue and red stand for would be helpful.

Thank you for the suggestion, labels have been added to indicate left and right trials for Figure 5 C/F and Figure 6A.

(14) Page 32: Legend does not include panel D.

Thank you for catching this, the corresponding caption has been added.

**Reviewer #2 (Recommendations for the authors):**
(1) The Introduction could perhaps set the central question in starker relief. What specifically do the authors mean by high- vs low-level control? As suggested by the cited studies, this has been a fraught issue in primate work for decades, and I think a finer-grained framing of alternative hypotheses would help set up the results. For example, would better performance at decoding joint angles than paw position be evidence for lower-level control? The clarity of the Introduction might also be improved if the facts and unknowns were broken down by species throughout.

We have tried to further improve the focus of the Introduction on the central question, clarify what we mean, and make clearer in the review of the literature which species a finding comes from.

The clarifying text from the introduction is quoted below:

Extensive motor mapping experiments in rodents have revealed that activating different parts of the sensorimotor cortex evokes movements of different body parts or different kinds of movements of the same body part, as it does in primates (for review, see (Harrison and Murphy 2014)). Yet it is unclear how the topography of stimulation-evoked movements relates to the roles of these areas during volitional actions. Perturbations during behavioral tasks in mice involving forelimb lever or reaching movements have provided a coarse-level understanding of how these areas contribute during behavior. Inactivations and lesions of M1 and S1 have shown that M1 is required for the execution of dexterous reach-to-grasp movements (Guo et al. 2015; Sauerbrei et al. 2020; Galiñanes et al. 2018; Wang et al. 2017; Whishaw et al. 1991; Whishaw 2000) and that S1 is essential for adapting learned movements to external perturbations of a joystick (Mathis et al. 2017). However, spinal cord projections from mouse M1 and S1 primarily target spinal interneurons rather than directly synapsing onto motor neurons (Gu et al. 2017; Ueno et al. 2018; Wang et al. 2017), suggesting cortical activity might play a more modulatory role. Further, stimulation of brainstem nuclei alone can evoke naturalistic forelimb actions, including realistic reaching movements involving coordinated flexion and extension of the proximal and distal limb (Esposito et al. 2014; Ruder et al. 2021; Yang et al. 2023). Taken together, these results have raised the question of what role mouse M1 and S1 play in the control of goal-directed forelimb movements.

One route to answering this question involves characterizing the signals present in mouse M1 and S1 during movement. If mouse M1 and S1 were to control only high-level aspects of forelimb movements, activity should be dominated by ‘abstract’ signals like target location and reflect little trial-to-trial variability in reach kinematics. If instead M1 and S1 control low-level movement features then activity should correlate strongly with forelimb joint angle kinematics and their trial-to-trial variation when reaching to different targets. While the presence of high- or low-level signals in a cortical area does not necessarily imply that they are causally responsible for these aspects of movement, characterizing what signals are present serves as a first step toward determining how these areas relate to movement.

(2) The kinematics and calcium traces appear to be highly stereotyped across trials. If the population encodes joint angles, would one expect to find correlations between the neural and kinematic residuals after subtraction of the time-varying means? Some additional analysis and/or discussion on this point would be helpful, especially as there are only two targets.

This is a great idea. As suggested, we implemented regression models on the residuals for each target in the new Figure 5-figure supplement 3. Figure 5-figure supplement 3 A and B show the performance when decoding the residuals for right trials and C and D show performance for left trials. Decoding remained well above chance, despite shrinking down due to predicting this relatively small within-target variation. This analysis supports our claims from the main regression models in Figure 5 and Figure 5-figure supplement 1-2, and also suggests that movements ipsilateral to the reaching limb (contralateral to the recording hemisphere) may be better encoded than movements contralateral to the reaching limb. We have added a reference to this additional residual analysis in the final paragraph of the decoding section of the Results section:

“Finally, we tested whether the ability to decode these many joint angles was a direct consequence of inter-joint correlations, and might not be indicative of the presence of “real” information about some of these joints. To do so, we fit partial correlation models that removed correlations between proximal and distal joints, or removed correlations of the joint angles with a high-level parameter – the overall distance of the paw centroid to the spout. Despite substantially lowering the behavioral variance, in each case the residuals could still be decoded from neural activity (Figure 5-figure supplement 2A-D). Similar decoding performance for M1-fl and S1-fl was obtained from models fit to decode single-trial residuals separately for left and right trials (Figure 5-figure supplement 3A-D), indicating that trial-to-trial variations on each basic movement were decodable from these populations.”

Along similar lines, binary classification is used to characterize cue-, lift-, and contact-responsive neurons. Is it possible to exploit trial-to-trial variation in the cue-lift and lift-contact latencies to extract the time-varying marginal effects of each event (e.g., using a GLM)?

For the detection of single-cell modulations by different events, we have elected to retain our simple statistical test to determine modulation; in our experience, encoding models typically involve a surprising number of steps to get them to do what you actually intend. We leave more extensive encoding model-style analysis to future work, currently in progress.

(3) The authors mention prior studies suggesting that the control of some forelimb tasks can be gradually transferred from the cortex to the subcortical centers. Have they performed the inactivation at different time points across learning, and if so, do they have evidence for a diminishing effect over time (e.g., blocking of both initiation and coordination early in training)? In addition, the effects of motor cortex inactivation are similar to, but slightly different from, effects shown in reaching tasks in prior studies. Some additional discussion on this point would be useful.

Our inactivation experiments in this study were intended to coarsely demonstrate the involvement of mouse forelimb sensorimotor cortex in our task. We have not performed the inactivations over learning and leave such experiments to future work.

We agree that a little more clarity relating our results to previous ones was warranted. Previous studies (Guo et al. 2015 and Galinanes et al. 2018) have demonstrated inactivation impacts on similar tasks, but for thoroughness we sought to show the same for our task as it varied from the pellet and motorized water spout tasks in both training time and target configurations. Our results are strongly in line with those of Galinanes et al. 2018 which used a fairly similar water spout target configuration. In the inactivation experiments of that paper, 3 out of 13 animals with initiation-triggered inactivations were able to initiate reaching within a time window similar to control trials. Additionally, a proportion of trials across multiple mice proceeded with little perturbation from the inactivations. This is consistent with our observation that M1-fl inactivations may either abolish movement initiation or allow movement initiation but impair task completion on a trial-by-trial and animal-to-animal basis. Further work is required to determine what factors influence these differential responses to inactivation and to determine how these effects differ across task variations (i.e., pellet vs water spout). We have added a brief description of these nuances to the text for clarity.

“These inactivations blocked the execution of the reach to grasp sequence, preventing the animal from making contact with the spout during the 3-second laser stimulation period (Figure 1F; 86.5% control trials with contact within 3 seconds of cue, 5.1% inactivation trials with contact, P < 10^-191^, Mann-Whitney U test, 2 mice, 495 stimulation trials). Interestingly, inactivation at the time of cue often did not prevent reach initiation (mouse 1: 54.7%, mouse 2: 34.2% of inactivation trials with lift within 3 seconds; 93.5%, 86.2% control trials). Yet the movement stalled once the paw and digits extended towards the spout, producing uncoordinated and unsuccessful reaching trajectories (Figure 1I, two representative datasets). Taken together, these results support the involvement of M1-fl in the water-reaching task and suggest that the strength of inactivation effects may depend on specific task details like training time or target configuration (c.f. Galinanes et al. 2018).”

Minor points(1) The rationale for the multiple comparisons procedure in identifying event-locked responses should be explained in more detail. If I understand correctly, the authors are not correcting for comparisons across ROIs, but instead control the family-wise error rate across brain regions and event types (dividing alpha by two or six). Why not instead control the false discovery rate across ROIs?

Thank you for pointing this out, it was confusing as written and we received a similar comment from Reviewer 1. We have fixed the wording now to make it clearer why we did this. We simply aimed to describe how many of the recorded neurons in each area were modulated by the task as a proxy for the engagement of these areas during the behavior, and to use this measure of modulation as a criterion for including the neuron in subsequent analysis. In other words, if the question had been “are any neurons in this area modulated by the task?” then correcting for the number of ROIs would be the correct method; but if the question is, “is this neuron probably modulated and therefore worth including in my decoder?” correcting for the number of ROIs will typically be much too conservative. Thus, we only sought to correct for the false discovery rate across events and targets for each ROI. We have added additional text in the methods to clarify these choices, below. Please also see response to (7) from Reviewer 1 above.

“Note that we did not correct for the number of ROIs tested for two reasons. First, the goal of this testing was to serve as a criterion for inclusion in subsequent decoding analyses, not to determine whether any neurons in the area at all were modulated; and second, correcting for the number of ROIs would bias comparison between areas if different numbers of ROIs were recorded in one area vs. the other.”

(2) It appears joint angles are treated as linear variables in the decoding analysis; is this correct? This seems reasonable as long as the range of motion is not too large, but the authors might briefly comment on the issue in the Methods.

Yes, all joint angles are treated as linear variables in the linear regression model. We observed empirically (as can be seen in Figure 3B and Figure 5B/F) that the joint angle variables were relatively constrained to specific ranges during the task, with no angles displaying substantial wrap-around during the reaching and grasping movements. It is true that use of nonlinear decoding would almost surely improve performance further. Future work could also compare decoding of joint angles with muscle forces, which correlate and which we made no effort to distinguish here. In this work, though, the demonstration of a substantial relationship between neural activity and kinematics already tells us that fine details of movement are present in the M1 and S1-fl populations, which is a critical fact to understand these areas and was not previously known. We now comment explicitly on this, as suggested.

“Joint angle or velocity kinematics were linearly interpolated from their original 6.66 ms to 10 ms and smoothed with a Gaussian (15 ms s.d.). These angular variables were then treated linearly in decoding analyses as their ranges were relatively constrained during the reaching and grasping movements; although the true relationships are likely nonlinear, this serves as a sufficient approximation to demonstrate the presence of a relationship between neural activity and kinematics.”

(3) Are the limb pose estimates mirrored along the mediolateral axis? Figures 1C and 2D appear to show reaches to the left spout on the animal's right.

Thank you for pointing out the ambiguity in the display of these data. The reach trajectories were not mirrored along the mediolateral axis, but they are displayed from the perspective of the behavioral imaging cameras as shown in Figure 1A. Thus the right target reaches (ipsilateral to the animal’s reaching arm) are on the left side of the camera image and the left target reaches (contralateral to the animal’s reaching arm) are on the right side of the image. We have clarified this in the figure captions.